# Quantifying 3′UTR length from scRNA-seq data reveals changes independent of gene expression

Mervin M. Fansler[1,2,3], Sibylle Mitschka [2,3] & Christine Mayr [1,2] ✉

Although more than half of all genes generate transcripts that differ in 3′UTR length, current analysis pipelines only quantify the amount but not the length of mRNA transcripts. 3′UTR length is determined by 3′ end cleavage sites (CS). We map CS in more than 200 primary human and mouse cell types and increase CS annotations relative to the GENCODE database by 40%. Approximately half of all CS are used in few cell types, revealing that most genes only have one or two major 3′ ends. We incorporate the CS annotations into a computational pipeline, called scUTRquant, for rapid, accurate, and simultaneous quantification of gene and 3′UTR isoform expression from single-cell RNA sequencing (scRNA-seq) data. When applying scUTRquant to data from 474 cell types and 2134 perturbations, we discover extensive 3′UTR length changes across cell types that are as widespread and coordinately regulated as gene expression changes but affect mostly different genes. Our data indicate that mRNA abundance and mRNA length are two largely independent axes of gene regulation that together determine the amount and spatial organization of protein synthesis.

In mRNAs, the 3′ untranslated region (3′UTR) is located between the coding sequence stop codon and the poly(A) tail. mRNA and 3′UTR length is determined by pre-mRNA cleavage and polyadenylation (CPA), which is initiated upon recognition of the polyadenylation signal (PAS) by the CPA machinery[1–3]. Approximately half of all human genes use alternative cleavage and polyadenylation (APA) to generate mRNA isoforms that differ in their 3′UTRs but encode the same protein[4–6]. In addition, ~25% of genes use intronic polyadenylation (IPA) signals to generate mRNA isoforms with alternative last exons, thus producing different protein isoforms[5,7–10]. APA is a widespread phenomenon, which is dysregulated in disease[1,2]. Alterations in 3′UTR length affect the presence of binding sites for microRNAs and RNA-binding proteins, and can regulate mRNA stability and translation[11,12]. More recently, 3′UTRs have emerged as important regulators of sub-cytoplasmic location of translation and mRNA-dependent co-translational protein complex assembly[13–22], reviewed in[3,23,24].

APA was initially posited to be a mode of gene expression regulation, where a switch in the 3′UTR isoform ratio results in changes of overall gene expression[11,12]. However, studies of transcriptome-wide APA reported that fewer than 20% of 3′UTR changes regulate mRNA or protein abundance. This suggested that mRNA abundance and 3′UTR length may be independent gene outputs[4,6,25–28]. Since these analyses were performed in fewer than ten cell types, it remains unclear whether the independence of gene and 3′UTR expression is a general phenomenon.

Whereas differential gene expression analysis is ubiquitously performed, the study of 3′UTR length is still very limited due to several technical roadblocks. Initial 3′UTR analysis methods required custom 3′ end sequencing protocols[4–6], which were not amenable for widespread use. Now, 3′-tag-based single-cell RNA sequencing (scRNA-seq) protocols can be used to quantify differential 3′UTRs[29–33]. However, most reads from 10x Genomics data do not span mRNA 3′ end cleavage

[1]Tri-Institutional Training Program in Computational Biology and Medicine, Weill Cornell Graduate College, New York, NY 10021, USA. [2]Cancer Biology and Genetics Program, Memorial Sloan Kettering Cancer Center, New York, NY 10065, USA. [3]These authors contributed equally: Mervin M. Fansler, Sibylle Mitschka. ✉e-mail: mayrc@mskcc.org

sites (CS), which prevents exact de novo mapping of mRNA 3′ ends. This limitation can be overcome by assigning reads to the closest known CS, obtained from CS databases[34,35]. These databases have grown substantially to contain ~300,000 CS for human protein-coding genes[35], and they present a highly complex landscape of alternative 3′UTRs.

We set out to reassess the CS landscape and generated a comprehensive CS atlas based on primary cells obtained from Microwell-seq (MWS) data of 206 human and mouse cell types[36,37]. Our MWS CS annotation was generated from 7 billion CS-spanning reads. It expands GENCODE CS annotations by 40% and enables classification into major and minor CS based on their usage across cell types, revealing that most genes have only one or two major 3′ ends. Moreover, we provide a fast and reliable workflow for simultaneous gene expression and 3′ UTR quantification from scRNA-seq data. We applied our quantification pipeline, called scUTRquant and a statistical testing package, called scUTRboot to scRNA-seq datasets covering 474 cell types and 2134 genetic perturbations and observed that changes in gene expression and in 3′UTR length occurred in different groups of genes, indicating that they largely represent independent regulatory events. We find that only about half of all gene regulatory events cause changes in mRNA abundance greater than 1.5-fold, whereas the other half affects mRNA length, which may impact the spatial control of protein synthesis. Together, we provide much needed resources and tools to enable profiling and mechanistic studies of 3′UTRs, from increasingly abundant scRNA-seq data.

## Results

### Mapping and characterization of mRNA 3′ end CS in 206 primary cell types

Typical 10x Genomics reads rarely contain untemplated adenosines, but approximately 35% of reads obtained from MWS span mRNA 3′ end CS (Fig. 1a)[36,37]. We analyzed MWS datasets from 104 murine and 102 human primary cell types, overall obtaining 7 billion CS-spanning reads to map mRNA 3′ ends at single-nucleotide resolution (Fig. 1b). Briefly, we removed poly(A) tails from the reads and mapped the remaining portions to the genome, and then filtered out low abundance reads as well as peaks derived from priming at genomic adenosine stretches. The resulting mRNA 3′ end CS were intersected with GENCODE annotations and classified into three groups, (i) common CS (GENCODE-annotated and observed in MWS data), (ii) MWS-only CS (not present in GENCODE), and (iii) GENCODE-only CS (annotated in GENCODE but not detected in MWS data). Our MWS CS annotation increases the number of CS relative to GENCODE by 40% (Fig. 1b). Two-thirds of the sites not found in GENCODE were detected in fewer than 10 cell types, suggesting that their absence in current annotations may indeed be attributable to their cell type-specific expression patterns (Fig. S1a). MWS-only CS were derived from tissues that span all developmental stages, including embryo, fetal, neonatal, and adult tissues. The largest fractions of MWS-only CS were obtained from adult tissues, such as omentum, pleura, and testis (Fig. S1b, c).

We assessed the quality of the MWS-only CS, by analyzing binding sites of the CPA machinery. Functional CS typically contain a PAS, an upstream cleavage factor (CF) I binding site and a downstream CFII binding site[3]. We also calculated APARENT2 scores which infer cleavage probability derived from a residual neural network model that was trained on data from a massively-parallel reporter assay[38,39]. Moreover, we analyzed PhastCons DNA sequence conservation surrounding the CS[40]. We observed that MWS-only CS have slightly weaker sequence contexts and lower predicted cleavage rates compared to common CS, but they show similar levels of sequence conservation (Fig. 1c−e, Fig. S1d−f) suggesting that they are proper, but weaker CS.

Next, we compared our MWS CS annotation with existing databases for individual genes. We found that the locations of both

common and MWS-only CS strongly agree with the CS locations in PolyA_DB V3.2 and PolyASite 2.0 (Fig. 1f−h, Fig. S2a)[34,35]. In contrast, GENCODE annotations are often incomplete and lack CS located closer to the stop codon, also called proximal CS (Fig. 1f, Fig. S2b). Most of the MWS-only CS were obtained from cell types that are absent in existing CS databases, as illustrated for human *ROCK1* (Fig. 1f). Although the *ROCK1* gene is expressed in 90 cell types, the additional distal CS was only detected in lung alveolar stem cells, lung macrophages, and cord blood hematopoietic stem cells (HSCs), thus making it a cell type-restricted site.

Analyzing CS usage across many cell types revealed that not all CS have similar usage rates, but instead can be broadly categorized into major and minor sites (Fig. 1g, Fig. S2c). For each CS, we calculated a CS usage score, representing the fraction of cell types that use a particular CS divided by the number of cell types that express the gene (Supplementary Data 1 and 2). We observed that nearly half of all CS are minor sites as they are used in less than 10% of cell types that express the corresponding gene (Fig. 1g, Fig. S2c). For 2674 genes, we observed that the most distal CS annotated in GENCODE are minor sites (Fig. 1f, Fig. S2b), indicating that many annotated 3′UTRs in GENCODE are misleadingly long. Importantly our analysis revealed that most expressed mRNAs have only one or two major 3′ ends. This is in contrast to the CS annotations in current CS databases that suggest that most genes contain more than five different 3′ ends (Fig. 1h, Fig. S2d).

### scUTRquant uses a truncated UTRome for fast and accurate gene and 3′UTR expression quantification

With this CS atlas in hand, we set out to develop a computational pipeline for gene and 3′UTR isoform quantification from raw scRNA-seq data. Our comprehensive reference CS atlas circumvents the need for de novo peak calling and its reliance on computationally intensive read mapping to a reference genome. Instead, we built a pipeline around the kallisto-bustools[41] workflow and implemented calibrations for resolving 3′ end isoforms. As reads mapping to different locations within 3′UTRs usually lack splice junctions, they cannot always be assigned unambiguously to different transcript isoforms. Therefore, we generated a truncated UTRome for pseudoalignment of 3′ end sequencing data, similar to Diag et al.[42] (Fig. 1b). We determined the cut-off for the truncation empirically and observed that more than 95% of 3′ end reads of tested reference genes map within 500 nucleotides (nt) upstream of CS (Fig. S3a). For closer spaced CS, we performed simulations to determine error rates for isoform quantification as a function of CS distance[43]. We observed that CS within 200 nt of each other could not be reliably quantified (Fig. S3b). Therefore, we merge their counts and assign them to the distal CS. We call this pipeline scUTRquant, which is available on our GitHub repository (https://github.com/Mayrlab/scUTRquant).

We tested the consistency of gene counts obtained from the truncated UTRome with standard processing, on several 10x Genomics scRNA-seq datasets[44]. Values from scUTRquant and Cell Ranger showed nearly perfect correlations, as evidenced by Spearman's rank correlation coefficients ($\rho$) exceeding 0.99 (Fig. S3c) for UMI counts per cell, and 0.92 for UMI counts per gene (Fig. 2a). Moreover, Louvain clustering results based on gene counts were similar (Fig. S3d, e). These results demonstrate that gene counts obtained from scUTRquant are consistent with the current gold standard analysis tool.

Next, we validated scUTRquant-derived 3′UTR isoform counts by comparing with values generated by bulk 3′ end sequencing methods. In FACS-sorted HSCs, we observed a strong correlation between scUTRquant values and bulk 3′-seq data (Fig. 2b, Spearman's $\rho = 0.86$)[45–47]. For embryonic stem cells (ESCs), the correlation was less strong (Fig. S3f, Spearman's $\rho = 0.70$), which may be caused by different cultivation conditions[48–51]. Nevertheless, we still consider the level of correlation between the current gold standard method and scUTRquant transcript counts as excellent, considering that the

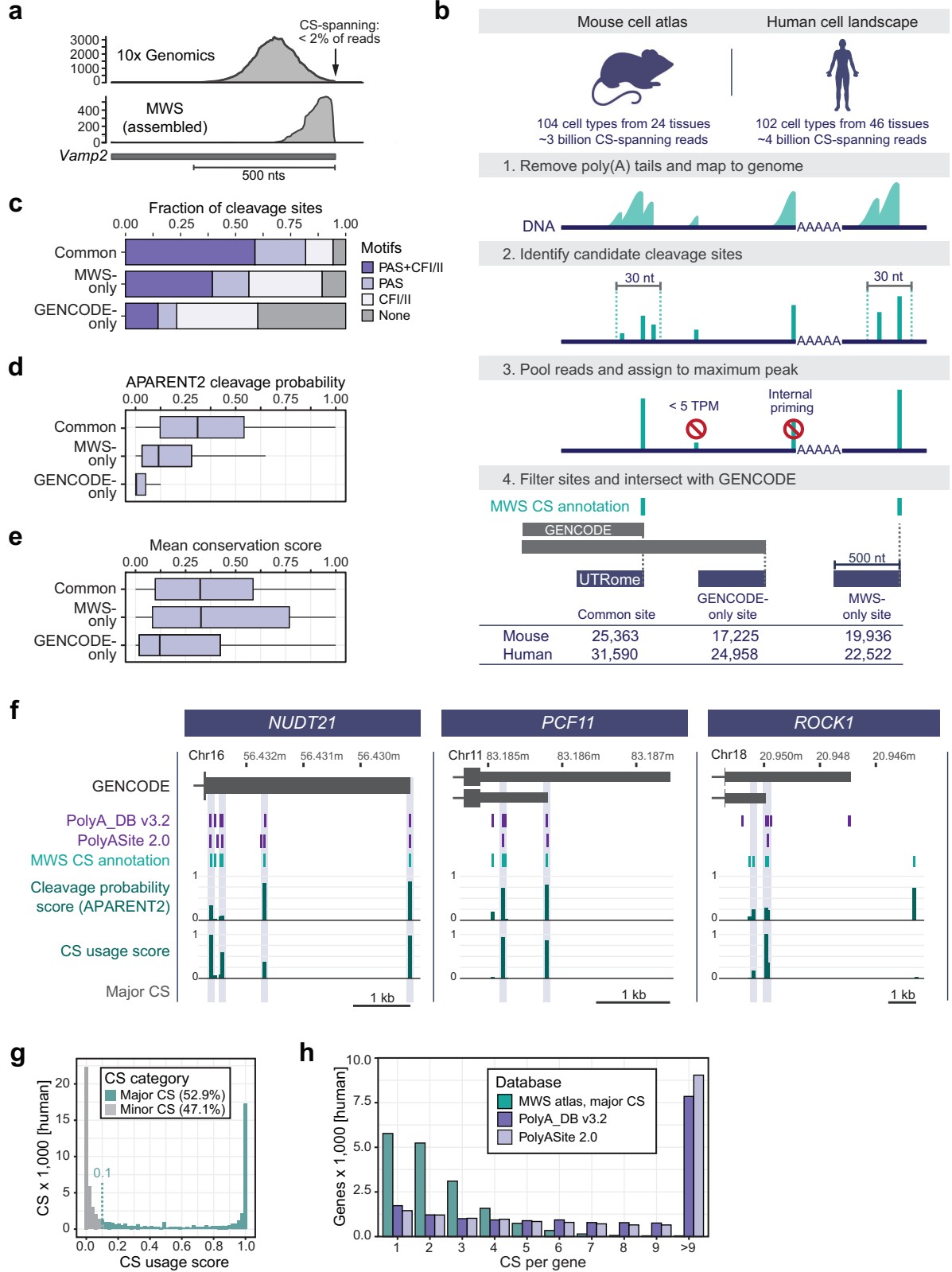

procedures were performed by different laboratories using vastly different methods.

Next, we assessed the reproducibility of 3′UTR isoform counts across biological replicates for both scRNA-seq and bulk 3′ end sequencing. We found that scRNA-seq samples exhibited a substantially stronger correlation than bulk 3′ end sequencing replicates (Fig. 2c, d, Fig. S3g, h)[45–51]. Importantly, scRNA-seq-derived biological

replicates were highly consistent, even when they were sequenced by different laboratories (Fig. 2d, Fig. S3h)[46,47,50,51]. We investigated potential reasons for the better reproducibility. As increased read depth or removal of PCR duplicates were not the cause, we speculate that higher data reproducibility in scRNA-seq may be due to standardization and automation of library preparation workflows. Together, these results reveal a better accuracy and substantially higher

**Fig. 1 | Mapping and characterization of mRNA 3′ end CS in 206 primary cell types. a** Read distribution at mRNA 3′ end CS from 10x Genomics compared with MWS data. Shown is the terminal exon of the mouse *Vamp2* gene. nts, nucleotides. **b** Schematic of CS annotation from MWS data and generation of a truncated UTRome for downstream gene and 3′UTR isoform quantification. **c** Motif distribution surrounding human CS (position 0). PAS (AWTAAA) in [−50,0], CFI binding site (TGTA) in [−100,0], CFII binding site (TKTKTK) in [0,50] for the indicated annotation categories. **d** Calculated APARENT2 cleavage probabilities for all human CS in a 30-nt window stratified by annotation category as shown in (**b**). Box shows interquartile range (IQR) with median and whiskers 1.5*IQR. **e** Mean

PhastCons score of 30 genomes in a 100-nt window centered on human CS, but excluding coding sequences. Box shows IQR with median and whiskers 1.5*IQR. **f** GENCODE transcript annotations depicting the last exons of the human *NUDT21*, *PCF11*, and *ROCK1* genes. Shown are chromosome coordinates (hg38), CS from existing PAS databases and our MWS CS annotation, together with APARENT2 cleavage probability scores and CS usage scores. Major CS are highlighted by the gray boxes. **g** CS usage score distribution for human CS that were identified by MWS. **h** Numbers of major MWS CS per gene compared to CS counts from two other databases. Shown are CS in all human protein coding genes.

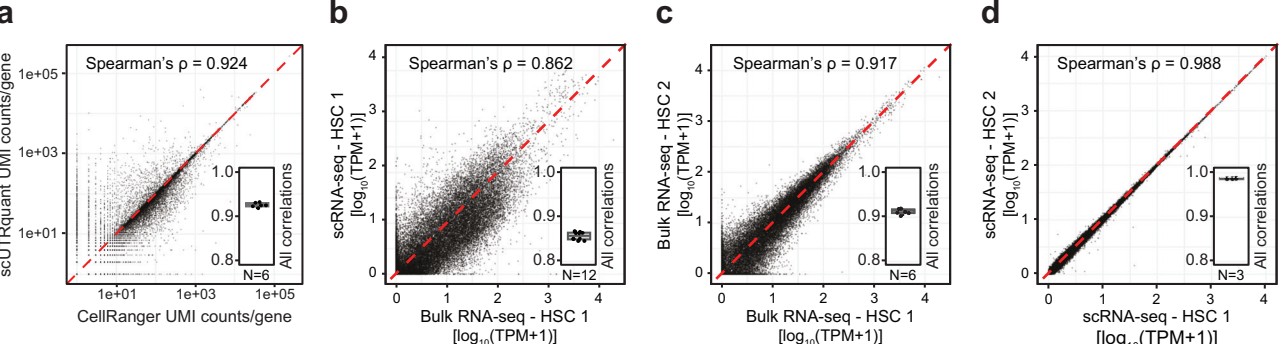

**Fig. 2 | scUTRquant-derived gene and 3′UTR expression is precise and accurate. a** Correlation of unique molecular identifiers (UMIs) per gene obtained by scUTR-quant and CellRanger for a mouse heart 10x Genomics demonstration dataset. Inset: Boxplot showing median and IQR of Spearman's ρ for six mouse 10x Genomics demonstration datasets. Whiskers indicate 1.5*IQR. **b** Correlation of 3′UTR isoform counts obtained from FACS-sorted HSCs comparing counts from bulk 3′ end sequencing methods with scUTRquant analysis of scRNA-seq data. Inset:

Spearman correlations from all ($N = 12$) pairs of HSC bulk and scRNA-seq replicates, shown as in (**a**). **c** Correlation of 3′UTR isoform counts between two replicates of bulk 3′ end sequencing for FACS-sorted HSCs is shown. Inset: Spearman correlations from all ($N = 6$) pairs of HSC bulk sequencing replicates, shown as in (**a**). **d** Correlation of scUTRquant 3′UTR isoform counts between two replicates of scRNA-seq of FACS-sorted HSCs. Inset: Spearman correlations from all ($N = 3$) pairs of HSC scRNA-seq replicates.

precision of 3′UTR isoform quantification from scRNA-seq data, suggesting that this method should become the new standard.

## scUTRquant and scUTRboot provide a workflow for 3′UTR analysis from scRNA-seq data

We conceived scUTRquant as part of a broader workflow for analyzing 3′UTR isoforms from scRNA-seq data (Fig. 3a). scUTRquant takes as input either raw scRNA-seq data (in FASTQ format) or output from other pipelines (in BAM format), together with a kallisto index from a truncated CS annotation file, called a UTRome. The comprehensive UTRomes for human and mouse are included as default options, comprising CS annotations for both intronic and last exon isoforms of all protein-coding genes. However, to provide flexibility for use with other organisms, we developed the Bioconductor package 'txcutr'[52] that can generate compatible indexes given any GFF or GTF transcriptome annotation (Methods). The scUTRquant pipeline outputs gene counts, 3′UTR isoform counts, or both, as well as quality control reports for all samples. Count matrices are formatted as Bioconductor 'SingleCellExperiment' objects that can be readily used for other scRNA-seq analysis applications. Moreover, previously established cell type annotations can be provided as input and scUTRquant will attach these as column data.

For identification of statistically significant changes in 3′UTR isoforms, we implemented a companion R package, called scUTRboot[53] (Methods). It provides a flexible set of non-parametric tests for changes in APA or IPA, and directional 3′UTR changes (shortening or lengthening).

## Classification of genes into single- and multi-UTR genes

Next, we used our scUTRquant pipeline to gain insights into broader patterns and characteristics of mRNA 3′UTR expression, from

scRNA-seq data. We analyzed single- or multi-UTR genes across 355 unique human cell types from the Tabula Sapiens dataset[54]. We classified a gene as multi-UTR based on the presence of at least two CS in the last exon, resulting in 3′UTR isoforms with relative expression of at least 10% of all 3′UTR counts in at least one cell type. Of 16,185 detected protein coding genes, 8056 expressed a single 3′UTR isoform, while 8129 genes were classified as multi-UTR genes, corresponding to 50% of human genes (Fig. S4a and Supplementary Data 3). Independently of single- or multi-UTR genes, we identified 4113 (25%) genes that generate IPA isoforms, thus changing the encoded protein (Fig. S4b and Supplementary Data 3).

Similarly, we processed scRNA-seq datasets comprising 119 mouse cell types obtained from Tabula Muris, brain, ESCs, and bone marrow[46,47,50,55,56]. Among the 16,195 expressed protein-coding genes, we classified 6766 (42%) as multi-UTR genes and we identified 1869 (12%) genes that generate IPA isoforms (Fig. S4c, d and Supplementary Data 4). Across mouse and human datasets, we observed that the majority of genes that are expressed in few cell types only generate one 3′UTR isoform, whereas most genes with broad expression patterns are classified as multi-UTR genes. Interestingly, true ubiquitously expressed genes (detected in more than 88% of cell types) are also more likely classified as single-UTR genes (Fig. S4e, f).

Among all genes classified as multi-UTR genes, we find that a fifth of them express more than one 3′UTR isoform at a rate of 10% or higher in only a few cell types (Fig. S4g–j). This suggests that some CS may predominantly produce lowly expressed isoforms. Moreover, we find that minor CS isoforms usually contribute only a small fraction of a gene's total expression across all tissue samples (6% and 4% in human and mouse, respectively). In contrast, major CS isoforms

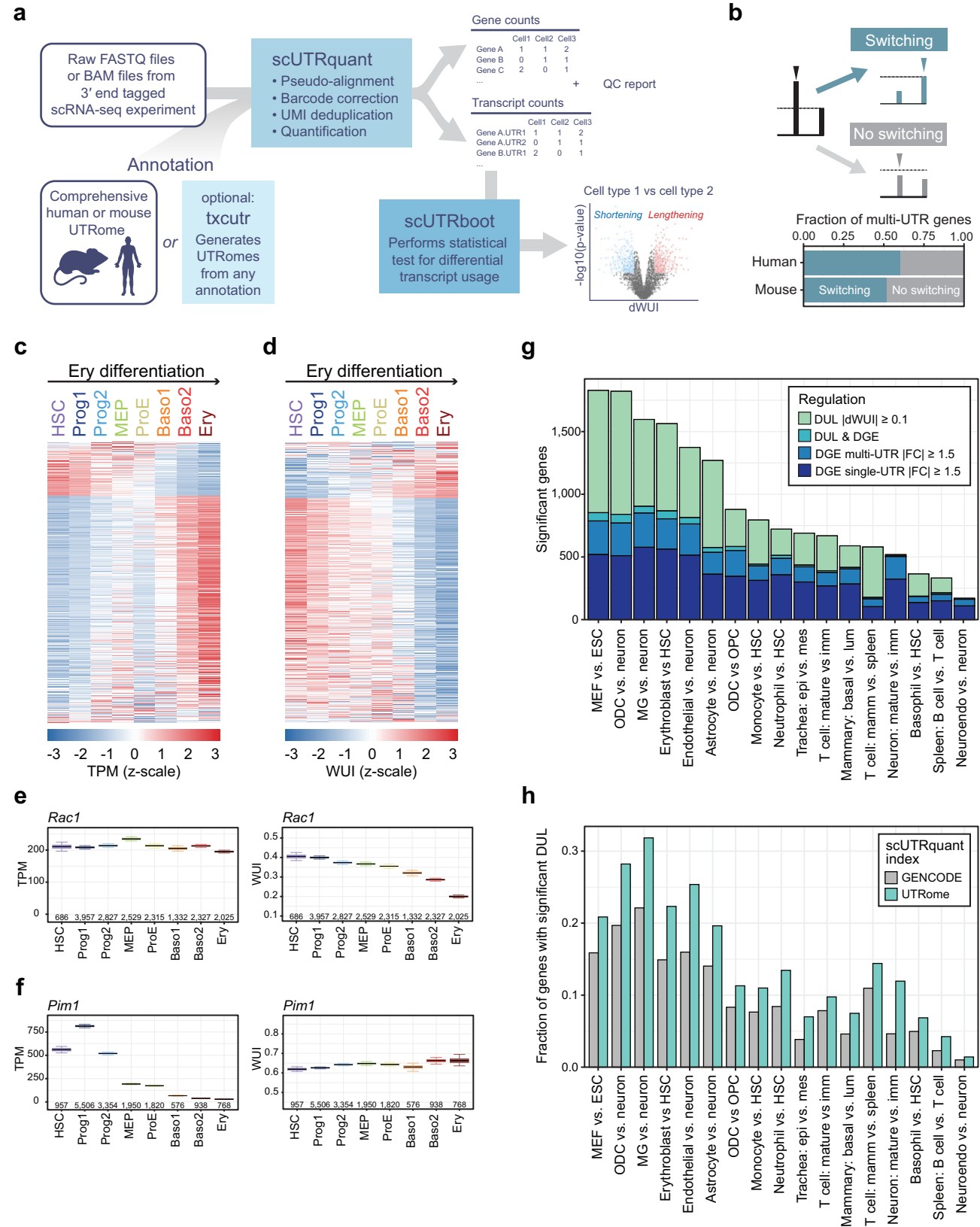

and those originating from common CS had the highest relative expression levels (Fig. S4k, l). Importantly, the majority of multi-UTR genes, 60% in human and 52% in mouse, switch between different dominant isoforms in at least one of the analyzed cell types (Fig. 3b), illustrating the highly dynamic expression pattern of 3'UTR isoforms.

## Coordinated 3'UTR length changes during differentiation

Next, we used the scUTRquant pipeline to re-analyze a published bone marrow dataset with the aim of identifying cell-type specific differences in 3'UTR length during red blood cell (Erythroblasts, Ery) differentiation from HSCs[46,47]. To classify 3'UTR changes into shortening or lengthening, we used a weighted UTR expression index (WUI)[57]. For

**Fig. 3 | scUTRquant analysis of diverse cell types demonstrates that changes in gene expression and 3′UTR length are independent gene regulatory events.** **a** scUTRquant pipeline schematic. Inputs are scRNA-seq raw data and a truncated CS annotation file. The txcutr tool generates truncated UTRomes for any genome. Outputs are quality control parameters and count matrices for gene and 3′UTR expression. The 3′UTR count output can be used as input for scUTRboot to identify significant differences across known cell types. **b** Fraction of all multi-UTR genes with dominant isoform switch in at least one cell type across 234 human and 82 mouse cell types. **c** Heatmap showing DGE for single- and multi-UTR genes between mouse HSC and Ery. DGE was tested with Welch *t*-test (|FC| > 1.5, q-value < 0.05), with 1059 genes increasing and 252 genes decreasing expression between Ery and HSC. HSC, hematopoietic stem cell, Prog1, progenitor cell type 1, Prog2, progenitor cell type 2, MEP, myeloid-erythroid progenitor, ProE, pro-erythroblast, Baso1, basophilic erythroblast 1, Baso2, basophilic erythroblast 2, Ery, polychromatic erythroblast. **d** As in (**c**), but shown is DUL for multi-UTR genes, calculated with scUTRboot's WUI bootstrap test (difference in WUI > 0.10, q-value < 0.05). Across the differentiation trajectory, 3′UTR lengthening and shortening is observed in 362 and 1275 genes, respectively. **e** Example gene (mouse *Rac1*) with significant DUL, but not significant DGE. Box shows median and IQR; whiskers are 95% confidence intervals for TPM (left) and WUI (right). Numbers of cells expressing the gene are indicated. **f** As in (**e**), but example gene (mouse *Pim1*) with significant DGE, but not significant DUL. **g**, As in (**c**) and (**d**), but pair-wise cell type comparisons were performed. Shown is significant DGE of single- and multi-UTR genes. For multi-UTR genes, DUL only or DUL and DGE is also shown. MEF, mouse embryonic fibroblast; ODC, oligodendrocyte; MG, microglia; OPC, oligodendrocyte precursor; epi, epithelial cell; mes, mesenchymal cell; imm, immature; lum, luminal cell; mamm, mammary; neuroendo, neuroendocrine. **h** As in (**g**), but the fraction of DUL genes relative to the number of expressed multi-UTR genes is shown, when using two different CS annotations as input for scUTRquant.

---

genes with two or more 3′UTRs, the isoform expression ratio is weighted based on the order of occurrence, with the shortest and longest isoforms being assigned weights of 0 and 1, respectively (Fig. S5a). The higher the WUI value of a gene, the more of its expression is derived from long 3′UTRs. For example, for genes with two 3′UTRs, the WUI represents the fraction of UMI counts that map to the longest 3′UTR isoform.

Along the Ery differentiation trajectory, we identified 1311 genes with differential gene expression (DGE) using a minimum fold-change of 1.5 as well as 1637 genes with differential 3′UTR length (DUL), considering WUI changes of at least 0.1 (Fig. 3c–f). Along the Ery differentiation trajectory, both DGE and DUL changes were gradual and coordinated, and they affected similar numbers of genes.

## Gene expression and 3′UTR length represent independent axes of gene regulation

To examine if changes in gene expression and 3′UTR length affect the same genes, we assessed pair-wise comparisons of HSC and Ery and identified 876 DGE and 873 DUL changes between the two cell types (Fig. S5b). Only 103 genes simultaneously changed both gene expression and 3′UTR length, indicating that, during differentiation, the majority of genes were affected by only one of these regulatory processes (Fig. 3g).

To examine the relationship between changes in gene expression and 3′UTR length more broadly, we determined DGE and DUL in an additional 16 differentiation and cell type sample pairs, including ESC, hematopoietic and neuronal cell types. In most pairwise comparisons, we observed similar numbers of gene expression and 3′UTR length changes (Fig. 3g). Importantly, the genes that changed their mRNA abundance or their 3′UTR length had little overlap, which ranged from 0–7.4%, considering both types of changes (Fig. 3g). The degree of overlap was consistent with statistical independence in 14/17 sample pairs (Supplementary Data 5). The overlap between DGE and DUL was still minimal, even when we lowered the cutoff for the expression fold change to 1.25 (Fig. S5c, Supplementary Data 5). To further exclude a bias caused by data thresholding, we calculated Pearson correlation coefficients across all pairwise comparisons. The average correlation coefficients were estimated to be less than 0.1, and 3/17 comparisons yielded no significant correlation between the gene expression and 3′UTR length variable (Fig. S5d, Supplementary Data 5). These data demonstrate that for most genes, expression and 3′UTR length changes indeed represent two independent axes of gene regulation (Supplementary Data 5).

## The MWS CS atlas increases detection of differential 3′UTR length by 40%

To investigate whether the MWS CS annotation can resolve more differential 3′UTR events, we calculated the number of DUL changes when using GENCODE annotations compared with using our UTRome.

When using the MWS CS annotation, we were able to test 20%-80% (median 70%) more multi-UTR genes and detected 110%-220% (median 150%) more differential 3′UTR events (Fig. S5e). When normalizing for the number of expressed multi-UTR genes that were analyzed, we still observed a median 40% higher fraction of genes with differential 3′UTR events compared to using GENCODE annotations only (Fig. 3h). This shows that our more comprehensive CS annotation substantially increases the number of genes with detectable and significant changes in 3′UTR expression across diverse cell types.

## 3′UTR analysis of a Perturb-seq dataset identifies previously unknown regulators of APA

We anticipate that the most common application for scUTRquant will be the quantification of 3′UTR isoforms across cell types. To demonstrate additional uses, we applied scUTRquant to a Perturb-seq dataset containing 2134 knockdown experiments for essential genes, to identify so far unknown regulators of APA[58]. To validate this approach, we plotted global APA and IPA changes after knockdown of known regulators. The scUTRquant analysis recapitulated previously published results, namely that knockdown of core CPA factors causes overall 3′UTR lengthening, whereas knockdown of CFI strongly induces 3′UTR shortening (Fig. S6a–h)[33,59,60]. Knockdown of the PAF complex also increased IPA, which is consistent with its known role as positive regulator of transcription elongation (Fig. S5a–h)[1]. The effect of each perturbation on global 3′UTR shortening or lengthening is reported in Supplementary Data 6.

To identify additional APA regulators, we calculated a z-scaled difference in WUI (dWUI) between each perturbation and the group of 97 non-targeting controls, followed by clustering on the perturbations and the response genes (Fig. 4a). We identified 18 perturbation clusters, which contain groups of factors, that when knocked down in K562 cells, caused similar patterns of 3′UTR shortening or lengthening in specific groups of response genes (Fig. 4a, b, Supplementary Data 7 and 8). Among the perturbation clusters, we identified several known APA regulators, including members of the CPA machinery as well as splicing, nuclear export, and transcription elongation factors. We also observed several clusters that contain members of large protein complexes, including CCT, nuclear exosome, proteasome, and the ribosome (Fig. 4a, b).

When performing gene ontology analysis with the set of 836 APA regulator genes compared with all analyzed 2057 essential genes, we found that "RNA binding" (GO:0003723) was the top-ranked term ($1.87 \times 10^{-28}$ FDR-corrected p-value). This result indicates that a large fraction of proteins encoded by the APA regulators can directly interact with mRNA. In contrast, we observed a significant de-enrichment of terms related to intracellular transport, DNA binding and actin binding (Supplementary Data 7). This suggests that knockdown experiments of essential genes related to these important functions do not elicit specific APA pattern changes. Taken together,

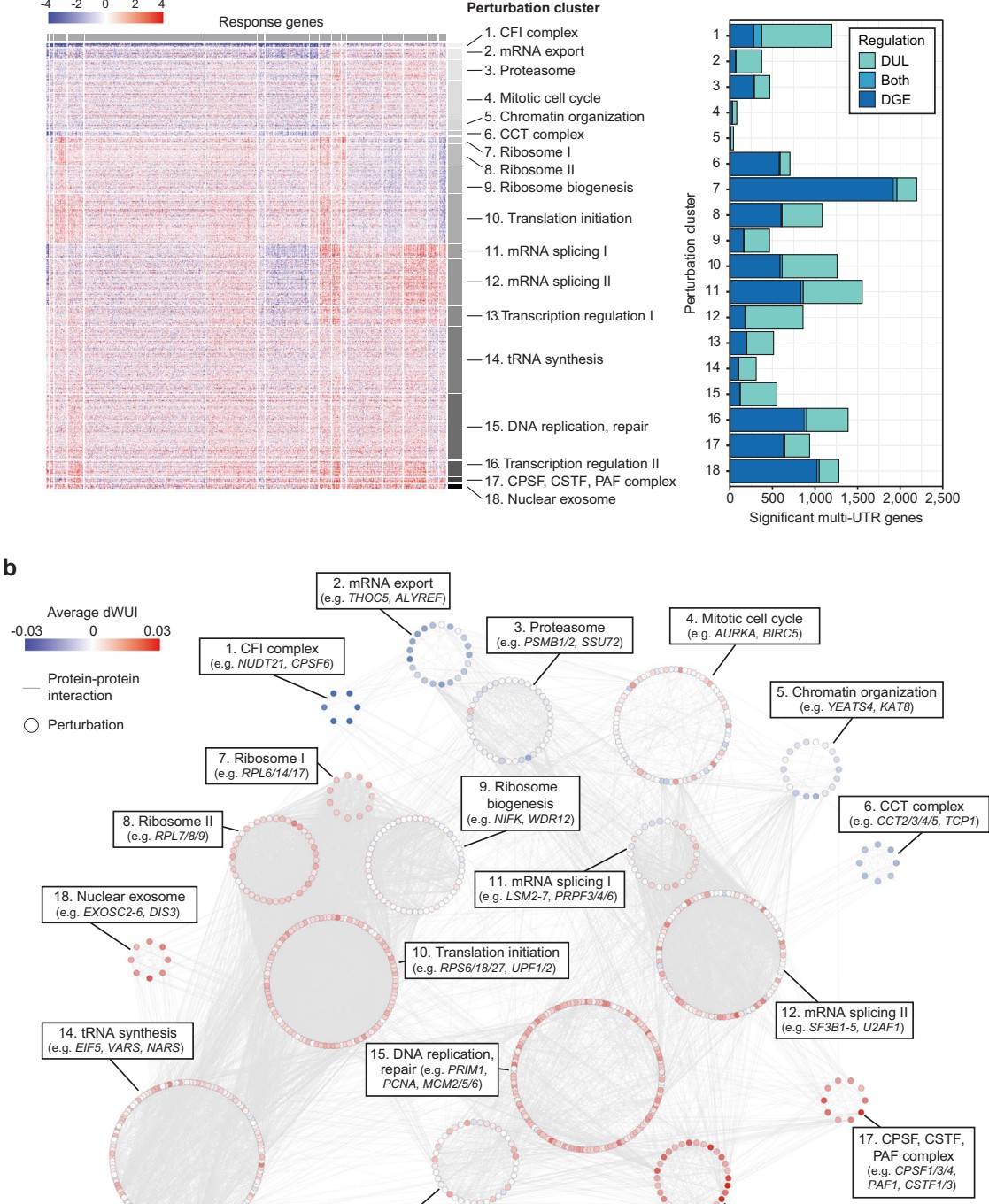

**Fig. 4 | Perturbation clusters that cause similar patterns of 3'UTR changes in specific groups of target genes. a** Clustering of essential gene perturbations on z-scaled dWUI values identifies 18 clusters with gene perturbations causing distinct 3'UTR expression patterns. Gene perturbations ordered by perturbation cluster are located on the vertical axis (836 genes) and response genes are presented on the horizontal axis (883 genes). Blue and red colors indicate shifts towards expression of shorter or longer 3'UTRs upon perturbation, respectively. **b** Members of 18 perturbation clusters from (**a**) are shown in a protein-protein

interaction network based on data from the STRING database. Nodes representing individual perturbations were arranged by cluster, and average dWUI values for each perturbation are indicated by node colors. **c** Bar plot showing the number of genes with DGE and DUL in each perturbation cluster (as in (**a**)). DGE was determined by two-sided Mann-Whitney test on all genes with a minimum expression of 5 TPM, showing a |FC| > 1.5 with q-value < 0.05. DUL was tested on multi-UTR genes with a minimum expression of 5 TPM and a q-value < 0.05.

despite originating from the same gene, steady-state expression regulation of 3′UTR isoforms can occur at various processing stages, including through posttranscriptional regulation.

To assess if the dWUI changes observed in each perturbation cluster are reproducible, we analyzed the 3′UTR isoform changes upon knockdown of the same factors in a different cell type (RPE1 cells)[58]. For all 18 clusters, we observed significant positive correlations between the cluster-average z-scaled dWUIs and the z-scaled dWUIs of the replicating perturbations, showing that the 3′UTR isoform changes correlated significantly (Fig. S6i). These results validate our experimental approach and indicate that the identified APA regulators induced similar 3′UTR isoform changes in two different cell systems.

As the majority of identified APA regulators were previously reported to be involved in gene expression regulation[58], our data suggests that gene and 3′UTR isoform expression are controlled by a shared set of factors (Fig. 4c). To identify whether specific complexes have a bias towards gene or 3′UTR regulation, we intersected DGE and DUL for each perturbation cluster. Whereas perturbing the core CPA machinery mostly affected gene expression, CFI knockdown was biased towards 3′UTR regulation (Fig. 4c). Surprisingly, single-UTR genes were not enriched among DGE events, indicating that the gene class alone does not predict the preferred mode of gene regulation (Fig. S6j). These results further suggest that multi-UTR genes can change either their expression or their 3′UTR length depending on the context.

### Reduction of ribosomal proteins causes widespread APA changes

To obtain deeper insights into APA regulation, we divided the changes induced by each perturbation cluster into 3′UTR shortening or lengthening (Fig. 5a and Supplementary Data 8). The strongest APA regulator was CFI, whose knockdown nearly exclusively induced 3′UTR shortening, as was reported previously for other contexts[33,59,60]. Perturbation of mRNA export factors also predominantly induced 3′UTR shortening (Fig. 5a), indicating that the presence of CFI or export factors promotes full-length mRNA isoform expression. In addition to CFI, perturbation of splicing factors caused the largest numbers of APA changes (Fig. 5a). This was followed by perturbation clusters that contain general transcription factors, translation initiation factors, and the ribosome (Fig. 5a). Although inhibition of translation is known to be associated with mRNA decay[61], it was surprising to find that inhibition of factors that target the mRNA region common to 3′UTR isoforms can differentially affect their expression. Together, we found that ~64% of analyzed multi-UTR genes (2032 genes) exhibited a significant DUL change in at least one cluster condition. In 522 genes, we recorded a switch of the dominantly expressed isoform, with some clusters causing more switching events than others (Fig. S7a).

Next, we characterized the extent of overlap in target genes that are regulated in the same or opposite direction across perturbation clusters (Fig. 5b). As expected, perturbation clusters that contain genes with related molecular functions showed the strongest overlap. For example, all four clusters with genes related to ribosome function induced 3′UTR shortening (or lengthening) of common target gene sets. The two splicing clusters also affected similar genes. Interestingly, we observed that the targets of CFI and mRNA export factors were highly overlapping (Fig. 5b), which suggests that these factors may be part of a common pathway. Moreover, whereas perturbation of splicing, transcription, and CPA factors induced 3′UTR lengthening of similar genes (Fig. 5b), perturbation of the ribosome caused 3′UTR shortening of this gene set (Fig. 5c).

### Most APA changes affect the shorter 3′UTR isoform

APA changes are commonly described as 3′UTR shortening or lengthening, but these patterns arise through various types of isoform changes (Fig. 5d). 3′UTR shortening can occur through exclusive upregulation of the shorter isoform, exclusive downregulation of the longer isoform or changes that affect both isoforms (Fig. 5d). Across all changes that affect genes with two 3′UTR isoforms, we observed that balanced changes, where both isoforms change in opposite directions, are not the dominant mode of regulation (Fig. 5d). Balanced regulation events occur most frequently in clusters likely involved in nuclear mRNA processing (Fig. S7b). Most often, the shorter 3′UTR isoform changes abundance (Fig. 5d). Interestingly, genes that were affected by many perturbation clusters had significantly weaker proximal CS (Fig. S7c–e). Taken together, these results suggest that isoform-specific 3′UTR transcript regulation is very common and that proximal CS are most important for dynamic regulation.

### Gene features of the coding sequence are frequently associated with APA

To better understand why specific groups of response genes are affected by particular perturbation clusters, we identified gene and mRNA features that correlate with 3′UTR changes, stratified by each perturbation cluster (Fig. 5e, Fig. S8). For example, a high percentage of sub-optimal codons promoted 3′UTR shortening in most perturbations (Fig. 5e, Fig. S8). In addition to codon optimality, we identified several additional features that are intrinsic to the gene or mRNA region common to both 3′UTR isoforms. These features include maximum intron length and coding region length (Fig. 5e, Fig. S8 and Supplementary Data 9).

We further observed that stronger distal PAS scores are associated with 3′UTR shortening in several perturbation clusters (Fig. 5e, Fig. S8). This result is not intuitive at first glance but may be explained by the fact that CS with the highest PAS quality metrics often employ auxiliary mechanisms to promote cleavage. For example, high-scoring PAS sites usually harbor NUDT21 binding sites, which allow binding of the CFIm complex, thus strongly enhancing pre-mRNA cleavage[26]. Hence, the knockdown of CFIm components (cluster 1) preferentially causes shortening in genes where the distal sites have NUDT21 binding sites, and these sites usually have high PAS scores. Similarly, the sequence context surrounding the CS could further improve isoform expression by coupling 3′ end cleavage and mRNA export[62,63]. Since long isoforms are more dependent on dedicated export pathways, the depletion of export factors may lead to preferential shortening in genes that usually employ these mechanisms.

Steady-state 3′UTR isoform expression levels are controlled by both mRNA production and degradation. We observed a significant correlation between estimated isoform half-life and isoform expression changes across many perturbation clusters, especially among those that contain proteins primarily involved in transcriptional processes (Fig. 5e, Fig. S8). Isoform half-life correlates significantly with perturbation clusters 1, 2, 11-13, and 15-18, which all contain factors involved in transcription and mRNA processing, suggesting that a large part of mRNA degradation occurs co-transcriptionally and during mRNA maturation/export[64,65]. These results reveal that mRNA production and degradation are linked and cannot be cleanly separated.

## Discussion

Our study presents a comprehensive annotation of human and mouse mRNA 3′ end CS using data from hundreds of primary cell types that span all major organs and developmental stages[36,37]. We performed rigorous quality control and filtering for internal priming and show that MWS CS are of high quality. By integrating our MWS-derived CS with available GENCODE annotations, we provide the most comprehensive CS catalog for human and mouse to date, extending current GENCODE CS annotations by 40%.

In addition to expanding CS annotations, our large-scale analysis on cell type-specific CS usage allowed us to categorize CS into major and minor sites. This revealed that most genes only have one or two major CS. As their locations within the mRNA are identical across cell

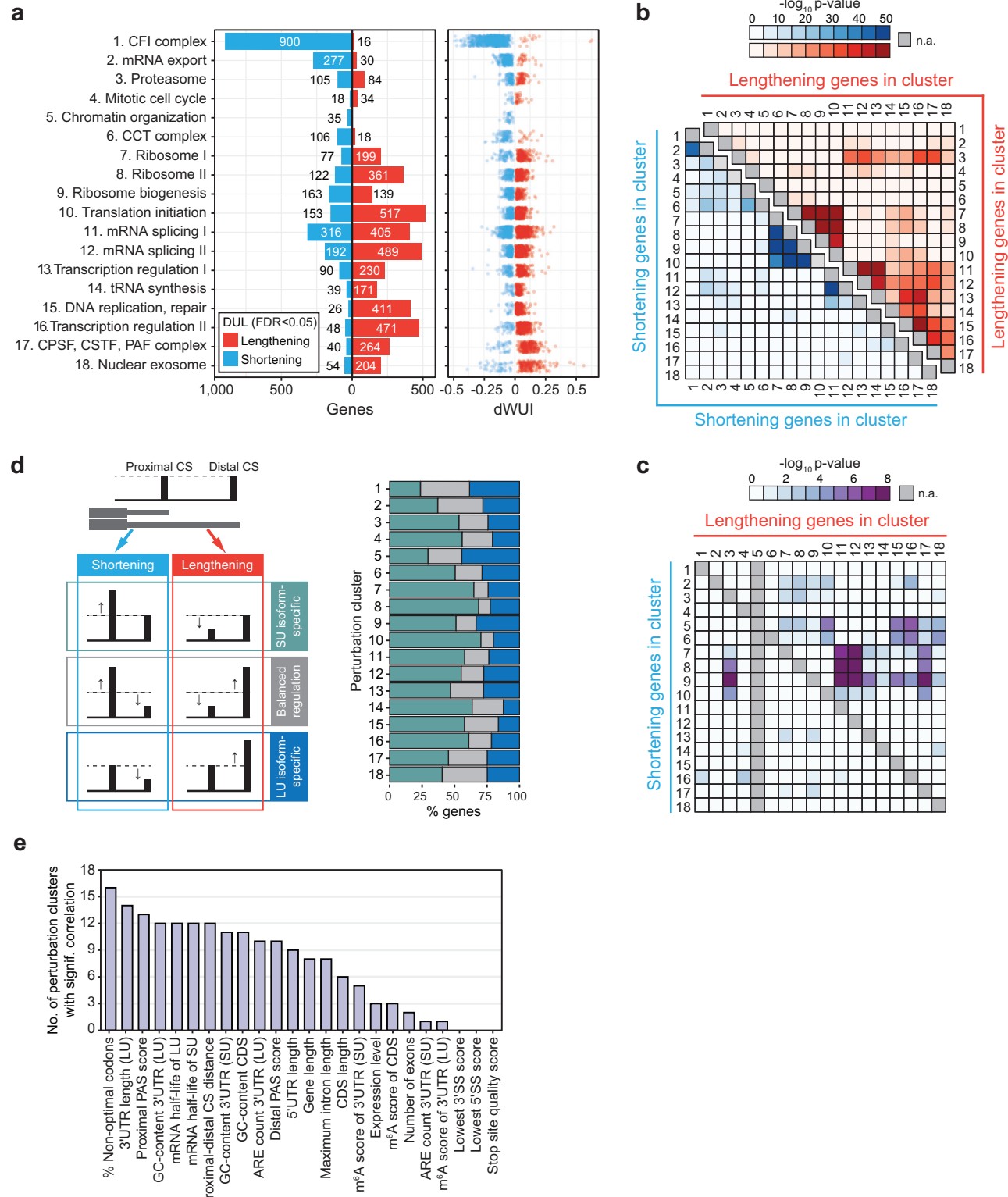

types, our data suggests that CS locations are intrinsic gene features. With these data, we would argue that the locations of major CS should be used to re-define 3′UTR boundaries in widely used transcriptome annotations. Within recent years, GENCODE and RefSeq databases have accumulated 3′UTR annotations and have included very long 3′ UTRs for many genes in human and mouse (Fig. 1f, Fig. S2b). Although these CS may be used in a few rare cell types, they represent minor CS that should not be used for the definition of 3′UTR length under most conditions. Our categorized CS annotation clarifies transcriptome-

wide mRNA 3′ end boundaries and increases our understanding of cell type-specific differences in mRNA 3′ ends.

In addition to the CS annotation, we developed an open-source computational pipeline, that makes it simple to quantify gene and 3′ UTR isoform expression from new or existing scRNA-seq data. By building on the kallisto-bustools workflow, we provide a fast and direct means of quantification that is compatible with most 3′-end tag scRNA-seq libraries. We established optimal parameters for reliable quantification and implemented these as default settings. With our pre-

**Fig. 5 | Regulatory logic of 3′UTR changes in 18 perturbation clusters. a** Left panel: Bar plot showing the number of genes with significant 3′UTR shortening (blue) or lengthening (red) in each perturbation cluster relative to 97 samples expressing non-targeting guide RNAs. DUL analysis was performed using a two-sided Mann-Whitney test with a q-value < 0.05. Right panel: Dot plot showing the average WUI difference for significant genes in the cluster. **b** Heatmap depicting the probability of significant gene overlap between perturbation clusters with synergistic regulation either in 3′UTR lengthening (upper right, red) or 3′UTR shortening (lower left, blue), calculated using a one-sided Fisher's exact test. **c** As in (**b**), but for antagonistic regulation of gene sets. **d** Left panel: Schematic diagram showing potential mechanisms leading to DUL. Regulation can either occur in a transcript-specific manner (affecting only one of the isoforms) or in a compensatory fashion (both isoforms change in a coordinated manner). Right panel: The mechanism of change, categorized as either short UTR (SU)-specific (green), coordinated (gray), or long UTR (LU)-specific (blue) is shown for the significant genes from (**a**). Coordinated change was defined as a relative shift of expression of at least 50% from one to the other isoform. **e** Gene and mRNA features that are significantly associated with 3′UTR changes (dWUI) observed in each perturbation cluster. Shown are the number of significant correlations with q-value < 0.05 for each feature. ARE, AU-rich element; CDS, coding sequence; SS, splice site. See also Fig. S8 and Supplementary Data 9.

defined CS annotation, scUTRquant quantifies a consistent set of 3′ UTR isoforms, making it easier to integrate datasets. Coupled with scUTRboot, significant differences in 3′UTRs across samples are identified, which facilitates the integration of 3′UTR quantification into standard scRNA-seq data analysis.

To demonstrate how scUTRquant can be used to gain new biological insights, we analyzed the global 3′UTR changes in a Perturb-seq data set that contains over 2000 knockdown experiments[58]. We established a comprehensive catalog of 3′UTR regulators that validated known factors and substantially expanded our knowledge of the mechanisms of APA regulation. For instance, in addition to factors of the CPA machinery, we observed that splicing has a substantial and widespread impact on expression of alternative 3′UTRs. Moreover, we identified the ribosome and translation initiation as major influencers of differential 3′UTR expression. While translation has long been recognized as an important regulator of mRNA decay[61], we provide evidence for translation-dependent differential turnover of mRNAs with alternative 3′UTRs. This may be surprising as the coding regions of alternative 3′UTR isoforms are identical. Our data suggest that 3′ UTR-bound RNA-binding proteins communicate with the translation environment to trigger translation-dependent mRNA decay. This may allow the ribosome to integrate signals from different parts of the mRNA, including codon usage from the coding sequence and 3′UTR-bound RNA-binding proteins.

We further applied the scUTRquant pipeline to 3′UTR analysis across 474 human and mouse cell types. We found that in 60% of human genes the dominant isoform switches in at least one cell type (Fig. 3b). By integrating changes in gene expression and in 3′UTR length across many cell types, our analysis revealed that mRNA abundance and 3′UTR length are two independent measures of gene output (Fig. 3g). Our results confirm previous observations that were obtained from a limited number of cell types[4,6,25–28]. Here, we demonstrate that only approximately 10% of differential 3′UTR events are associated with changes in gene expression, which indicates that in most cases, APA is not a mechanism for gene expression regulation. Importantly, we revealed that only approximately half of all changes in gene output affect mRNA abundance, whereas the other half affect 3′ UTR length and, therefore, control the presence or absence of regulatory motifs in mRNAs.

What is the reason why most significant changes in 3′UTR isoform expression are not associated with significant gene expression changes? We observed that the majority of 3′UTR isoform abundance changes affect the less abundant isoform (Fig. S5f). As a result, even a two- or three-fold change in isoform abundance does not change overall gene expression by more than 1.5-fold. Whereas 3′UTR isoform changes may not alter overall mRNA level of a gene, the 3′UTR changes still have important consequences for individual isoforms.

Importantly, 3′UTRs are known as major regulators of mRNA localization[23,24], even in non-neuronal cell types and cell lines[15,21,66–68]. It was shown recently that mRNA architecture features, including mRNA and 3′UTR length, correlated strongly with subcytoplasmic mRNA distribution[21]. Moreover, 3′UTRs have emerged as important regulators of mRNA-dependent assembly of proteins complexes[13,14,18,19,22]. Taken together, our large-scale analysis revealed that about half of all gene regulatory events captured in scRNA-seq data go undetected when using standard gene expression analysis pipelines. Rather than controlling the abundance of transcripts, these changes may instead predominantly modulate where in the cytoplasm an mRNA isoform is translated.

## Methods

### Cell type-specific identification of mRNA 3′ ends from primary cells using MWS data

**CS identification.** FASTQ files of MWS data from the Mouse Cell Atlas v1.1[36] (GEO:GSE108097) and the Human Cell Landscape[37] (GEO:GSE134355) were downloaded and then assembled using PEAR v0.9.6 with settings '-n 75 -p 0.0001'. Cell and UMI barcodes were extracted from assembled reads and placed into read headers using umi_tools v1.1.2; remaining poly-T regions at the 5′ end of assembled reads were trimmed using cutadapt v3.5 with arguments '--front = ' T{100}' -g = 'T{12}' -n 10 -e 0', retaining only sequences with minimum length of 21 nts. Reads were aligned with HISAT v2.2.1 to the mm10 and hg38 genomes, respectively. Cell type annotations were used to demultiplex sample-level BAMs to cell-type-level for each dataset. Per cell type strand-specific coverage at the 5′ ends of aligned reads was computed using the 'genomecov -dz −5' command of BEDTools v2.30. Per cell type, all entries within 30 nt radius were merged to the local mode and retained when > 5 reads per million. Cell type coverages per strand were subsequently summed with GNU's datamash v1.7, and then a final pass of merging to the local mode within a radius of 30 nt was performed to harmonize minor variations in cell-type-level CS identification. The resulting sites were considered as CS candidates (mouse: $N = 170{,}617$; human: $N = 150{,}191$).

**CS filtering.** Candidate CS were intersected with 40 nt intervals centered at 3′ ends of GENCODE vM25 and v39 transcripts with positively identified 3′ ends (excluding tag 'mRNA_end_NF'), respectively. Intersecting sites (mouse: $N = 27{,}872$; human: $N = 31{,}026$) were classified as "validated"; non-intersecting sites were subsequently intersected with 40 nt intervals centered at cluster centers in the PolyASite v2.0 *Mus musculus* and *Homo sapiens* atlases using all clusters surpassing a 3 TPM threshold[35]. Intersecting sites (mouse: $N = 30{,}020$; human: $N = 29{,}897$) were classified as "supported"; non-intersecting sites were filtered through cleanUpdTSeq v1.32 with maximum posterior probability of 0.0001 of being an internal priming site[69]. Passing sites (mouse: $N = 22{,}824$; human: $N = 18{,}055$) were classified as "likely". The union of "supported" and "likely" CS was formed and each site was annotated according to the GENCODE vM25 and v39 annotations, with one of the ordered labels: "three_prime_UTR", "five_prime_UTR", "exon", "intron", "extended_five_prime_UTR", "extended_three_prime_UTR", or "intergenic", where the existing 5′ ends of transcripts were extended 1 kb upstream and existing 3′ ends of transcripts were extended 5 kb downstream.

**Generation of the MWS CS annotation and transcriptome truncation.** The GENCODE vM25 and v39 annotations were filtered for protein-coding transcripts with known 3′ ends. CS with a

"three_prime_UTR" label were intersected with these transcripts and new transcript versions ending at the CS were generated ("upstream"). All protein-coding transcripts with known 3′ ends were extended by 5 kb downstream, intersected with the "extended_three_prime_UTR" set of CS, and new transcript versions ending at the CS were generated ("downstream"). All transcripts (GENCODE, upstream, downstream) were truncated to include 500 nts from their 3′ end. Truncated transcripts with fewer than 50 nts difference were reduced to a single representative copy, with prioritization for downstream sites. The collection of remaining truncated transcripts was exported to GTF and the corresponding sequences to FASTA. This expanded the GENCODE annotations with 19,936 and 22,522 additional 3′UTR isoforms, respectively. This annotation is called MWS CS annotation. The GTF and BED files are deposited on figshare (https://figshare.com/s/0709e2551cc1ee4c4941).

Complete pipelines are deposited at https://github.com/Mayrlab/hcl-utrome (https://doi.org/10.5281/zenodo.8118411) and https://github.com/Mayrlab/mca-utrome (https://doi.org/10.5281/zenodo.8118416).

### Characterization of CS in the MWS CS annotation

**Sequence motifs surrounding CS.** DNA sequence in a window of 1000 nt centered at each CS was extracted. For each motif, the center positions of all occurrences were determined across all sequences. Smoothed density is computed with ggplot's 'stat_density', with global mode scaled to 1.

**APARENT2 cleavage probabilities.** DNA sequence in a window of 205 nt centered at each CS in the MWS CS annotation was extracted and used as input to APARENT-ResNet v1.0.2[39]. Cleavage probability for a site was computed as the sum of probabilities obtained from APARENT-ResNet in the 30 nt window centered at the CS.

**DNA sequence conservation scores.** Conservation scores were extracted in 100 nt windows centered at CS, with annotated coding sequence regions excluded, using the Bioconductor package 'GenomicScores' v2.10.0 with databases "phastCons30way.UCSC.hg38" and "phastCons60way.UCSC.mm10". Means were computed for each set of scores from the window.

**Comparisons with existing CS databases.** When counting the number of CS per gene we restricted the analysis to protein-coding genes. When analyzing for consistency with previous annotations, we calculated the strand-specific distance of each human MWS CS to the closest CS annotated in two other CS databases[34,35]. For this purpose, we performed liftover of CS annotated in PolyA_DB v3.2 to match the hg38 reference genome. When comparing to the PolyASite 2.0 database, we consider a MWS CS to have a distance of 0 nt relative to the closest PolyASite 2.0 CS if it falls within any CS of the annotated CS cluster. Plots of example genes included only clusters from PolyASite 2.0 with a minimum TPM of 1 or relative expression ratio above 0.05. PolyA_DB v3.2 CS were similarly filtered but using a 1 RPM threshold. GENCODE (human v39; mouse vM25) were filtered to exclude transcripts denoted with "mRNA_end_NF".

**The CS usage score classifies CS into major and minor sites.** The CS usage score is the fraction of cell types that use a CS in the data from MWS divided by the number of cell types that express the gene. If a CS is used in <10 cell types where the gene is expressed, it is considered a minor CS. All CS together with their CS usage scores are reported in Supplementary Data 1 and 2 (for human and mouse CS annotations).

### Definition of scUTRquant parameters

**Empirical distributions of 10x Genomics peak width.** A set of 56 peaks located at the 3′ ends of transcripts was manually curated by examining the genomic alignments of 10x Genomics Chromium v2 samples from the Tabula Muris dataset[55] (GEO:GSM3040890-GSM3040917). Peaks were selected for absence of splice sites, potential internal priming sites (A-rich regions), and nearby alternative CS in the immediate 800 nts upstream of the annotated CS. The coverage of 5′ ends of reads was extracted with the 'bedtools genomecov −5' command for each sample from the Tabula Muris dataset and the distance from the annotated CS of the corresponding transcript was computed. For each sample, the 95th percentile for distance from the 3′ end across all genes was computed. Additionally, for each gene, the 95th percentile for distance from the 3′ end across all samples was computed (Fig. S3a). Analysis code is available at https://github.com/Mayrlab/tmuris-peaks (https://doi.org/10.5281/zenodo.10895191).

**Kallisto transcript quantification resolution.** To resolve CS nearer than 500 nt apart we enabled the expectation maximization algorithm implemented in kallisto-bustools to proportionally assign ambiguous reads[43]. As 3′ end sequencing data violate the assumption of uniformly distributed reads used in the implementation[43], we investigated to what extent overlapping isoforms might induce quantification errors. We performed simulations to determine the error-rate as a function of CS distance.

The sequence of the Ensembl transcript Rac1-201 (ENSMUST00000080537) was used as the basis for a two-isoform transcript expression simulation. The first simulated isoform ("distal") used the annotated 3′ end; the second ("proximal") was created by removing specified intervals from the 3′ end. For each round of simulation, samples of read distances from the 3′ end of each transcript were generated according to a discretized gamma distribution with mean 300 and standard deviation of 100. Reads of 100 nts were generated using the respective transcript sequences and the randomly sampled positions. The 'kallisto quant' command was used to estimate transcript abundance, using the parameters '--single -l1 -s1 --fr-stranded --pseudobam' and truncated versions of the transcripts as index. Relative error for each transcript was computed using estimated and true abundances. A parameter sweep was performed with all combinations of the following parameters: (a) CS distances between [50-700] with 50 nt steps; (b) truncated transcript lengths [350−600] with 50 nt steps; (c) proximal counts {50,100}; (d) distal counts {50,100}. Each parameter combination was simulated for 10 replicates. Final resolution was selected based on mean relative errors approaching zero. We concluded that CS within 200 nt of each other cannot be reliably discriminated when quantifying (Fig. S3b), therefore, we configured the scUTRquant pipeline to merge their counts and assign them to the distal CS. Analysis code available at https://github.com/Mayrlab/kallisto-overlap (https://doi.org/10.5281/zenodo.10895237).

**Kallisto customization and scUTRquant settings.** The 'kallisto bus' command of kallisto version 0.46.2 was extended to support strand-specific pseudoalignment for both FASTQ and BAM input files. It is available at https://github.com/mfansler/kallisto/releases/tag/v0.46.2sq (https://doi.org/10.5281/zenodo.10902020). All 10x Genomics 3′ end datasets were pseudoaligned with 'kallisto bus --fr-stranded'. Cell barcodes for the corresponding technology version (v2 or v3) were used as whitelists for the 'bustools correct' command. Truncated isoforms in the same gene with 3′ ends nearer than 200 nts apart (mouse: $N = 18{,}057$; human: $N = 29{,}574$) were merged in the 'bustools count' step.

### txcutr generates truncated UTRomes for any genome annotation as input for scUTRquant

The default CS annotation for the scUTRquant pipeline is the human or mouse UTRome which contains the MWS CS annotation. Additional truncated UTRomes for any genome annotation can be created with the Bioconductor 'txcutr' package (https://doi.org/10.18129/B9.bioc.

txcutr) and used as CS annotation input for scUTRquant. The Bioconductor 'txcutr' package generates truncated GTF annotations, FASTA sequences, and merges tables[52].

To investigate how many additional differential 3′UTR events can be detected when using the mouse MWS CS annotation compared with GENCODE vM25, we used 'txcutr' v0.99.0 (equivalent to Bioconductor version 1.0.0). This index was generated with a 500 nt truncation length and a merge distance of 200 nts. In brief, the GENCODE vM25 annotation was first pre-filtered with an AWK script to remove any entries with the 'mRNA_end_NF' tag (indicating unvalidated 3′ ends) and restricted to protein-coding transcripts. The txcutr method 'truncateTxome' clipped all transcripts longer than the specified length, anchored at the 3′ end, intersected the truncated transcripts with the child exons of that transcript, and then redefined the genomic range of the gene to the union of all child transcripts. Transcripts that were identical after truncation were deduplicated to retain only one representative copy, which was annotated with the transcript ID of the transcript with lexicographical priority. The resulting TxDb object was then exported as a GTF and a FASTA file using txcutr's 'exportGTF' and 'exportFASTA' methods. Finally, a merge table was generated with txcutr's 'generateMergeTable' by further truncating transcripts to the specified merge distance, anchored at the 3′ end, intersecting within the parent gene, and recording the most downstream transcript with which each intersects. Additional specification and implementation details are found in the txcutr documentation. The generation of this index is reproducible from the Snakemake pipeline available at https://github.com/Mayrlab/txcutr-db (https://doi.org/10.5281/zenodo.8118405).

### Validation of scUTRquant-derived gene and 3′UTR isoform counts

**CellRanger and scUTRquant UMI count correlations.** To test the accuracy of gene counts obtained from the truncated UTRome, we compared gene expression values calculated with scUTRquant and CellRanger on six 10x Genomics 3′ end mouse demonstration datasets, available as FASTQ files from the 10x Genomics website ('heart_1k_v2', 'heart_1k_v3', 'heart_10k_v3', 'neuron_1k_v2', 'neuron_1k_v3', and 'neuron_10k_v3'). They were processed through the scUTRquant pipeline using the 'utrome_mm10_v2' target with default settings. The corresponding filtered HDF5 UMI counts from CellRanger 3.0.0 were also downloaded and loaded as SingleCellExperiment objects in R[44]. For each dataset, only cells (or genes) present in both the CellRanger and scUTRquant results were plotted and used to compute Spearman correlations.

Similarly, three 10x Genomics 3′ end human demonstration datasets ('pbmc_1k_v2', 'pbmc_1k_v3', and 'pbmc_10k_v3') were processed through the scUTRquant pipeline using the 'utrome_hg38_v1' target with default settings. Comparisons were performed against CellRanger UMI counts in the same manner as above.

The scripts and input files needed to download and run these datasets were incorporated into the scUTRquant pipeline as examples that users can run following the pipeline documentation.

**CellRanger and scUTRquant clustering comparisons using gene counts.** For each 10x Genomics dataset, the CellRanger and scUTRquant counts were filtered to common cells. Clustering was performed following Amezquita et al.[70]. In brief, size factors were computed with the 'computeSumFactors' from the 'scran' Bioconductor package[71], and then used to compute normalized log counts. The top 1000 high-variance genes were used to compute the first 20 principal components. Louvain clustering was performed on the cells in this reduced representation. The Adjusted RAND Index (ARI) between the CellRanger and scUTRquant clusters was computed using the 'aricode' R package.

### Comparison of 3′UTR isoform counts obtained by scUTRquant with bulk 3′ end sequencing methods.
For scRNA-seq data, we used mouse datasets from FACS-sorted HSCs (LSK samples) subsetting to only include cells given annotations in Wolf et al.,[46,47] (GEO:GSM2877127-GSM2877132) and from ESCs[50,51] (GEO:GSM3629847, GSM3629848, GSM4694997). The UMI counts were summarized to TPM per sample by aggregating counts across all cells in each sample and normalizing to UMIs per million. Bulk 3′-seq datasets on the same cell types were obtained. For FACS-sorted HSC samples[45] (ArrayExpress, E-MTAB-7391), 3′UTR isoforms were quantified by pseudoalignment of read 2 using the UTRome annotation and 'kallisto quant'. For bulk ESC datasets[48,49], TPM values and CS locations were obtained from the PolyASite v2.0 database[35] and intersected with the UTRome annotation using a 50 nt interval around cluster centers. TPM values between the samples were compared and Spearman's correlation coefficients were calculated.

### Application of scUTRquant to scRNA-seq data from 474 human and mouse cell types

**Classification of single- and multi-UTR genes from 119 mouse cell types.** Samples from Tabula Muris (GEO:GSM3040890-GSM3040917), ESC (GEO:GSM3629847, GSM3629848), bone marrow (GEO:GSM2877127-GSM2877132), and brain (GEO:GSM3722100-GSM3722115) datasets[46,47,50,55,56] were quantified for 3′UTR isoform expression following the default settings of scUTRquant with the 'utrome_mm10_v2' target and cells were annotated with published cell type annotations. Cell type labels for bone marrow cell types were obtained by combining publicly available transcriptome and proteome information for erythroblast differentiation[47,72]. Cells not previously annotated in published analyses were excluded.

All datasets were merged into one SingleCellExperiment object and counts were size-factor normalized using the 'computeSumFactors' method from Bioconductor package 'scran'[71]. UMI counts were aggregated by cell type and the percentage of isoform expression per gene was computed, excluding isoforms whose 3′ ends were located within a GENCODE-annotated intron of the corresponding gene. For each gene, the number of isoforms with at least 10% expression in at least one cell type were counted. Genes with two or more such isoforms were classified as multi-UTR genes; otherwise, they were classified as single-UTR genes. To identify genes that generate intronic polyadenylation (IPA) isoforms, all mRNA 3′ ends of a transcription unit were included. IPA isoforms were counted if they contained at least 10% of reads of a gene in at least one cell type. The data on mouse single-, multi-UTR, and IPA genes together with the 3′UTR length are reported in Supplementary Data 4. The Snakemake pipeline for classification is available at https://github.com/Mayrlab/atlas-mm (https://doi.org/10.5281/zenodo.10895352).

**Classification of single- and multi-UTR genes from 355 human cell types.** 10x Genomics samples from Tabula Sapiens[54] were downloaded in BAM format and processed similarly to the mouse data, but using the scUTRquant 'utrome_hg38_v1' target. The data on human single, multi-UTR, and IPA genes together with the 3′UTR length of each isoform are reported in Supplementary Data 3. The Snakemake pipeline for classification is available at https://github.com/Mayrlab/atlas-hs (https://doi.org/10.5281/zenodo.10895337).

**Analyzing CS usage as a function of transcript expression.** Cell type samples of human and mouse were subsetted to contain a minimum of 200 cells, resulting in 234 human and 82 mouse samples. Transcripts were filtered to those where all reads in the quantification window coud be unambiguously assigned to one CS (no merging) and total gene expression was > 5 TPM (excluding IPA). Isoforms were grouped by CS annotation category (common/ MWS-only/ GENCODE-only) or

CS usage category (major/ minor) as indicated in Supplementary Data 1 and 2. The isoform expression fractions for each CS transcript were calculated as TPM of isoform/total TPM of gene and the median across all cell type samples was plotted.

**Analysis of isoform switching.** The highest expressed 3′UTR isoform was identified for each multi-UTR gene with total expression > 5 TPM (excluding IPA). Only cell types that contained a minimum of 200 cells were analyzed (234 human and 82 mouse cell types, respectively). A gene was designated as switching if it had at least one different dominant isoform in at least one cell type.

**Identification of differential 3′UTR isoform expression between samples using scUTRboot**

To identify statistically significant changes among 3′UTR isoforms across samples, we developed a companion R package, called 'scUTRboot'[53]. It provides a flexible set of non-parametric testing procedures to test for changes in 3′UTR isoform or IPA isoform expression, directional changes (3′UTR shortening/lengthening) or focus on ratio changes in specific classes of isoforms. The scUTRboot R package is deposited at https://github.com/mfansler/scutrboot (https://doi.org/10.5281/zenodo.8057843).

**Two-sample bootstrap test with scUTRboot.** scUTRboot implements two-sample hypothesis testing with a bootstrap strategy for estimating p-values. The 'twoSampleTest' function implements three general modes of tests based on the statistic computed across the samples: a UTR expression Index (UI), a Weighted UTR expression Index (WUI), and a Wasserstein Distance (WD), also called the Earth Mover's Distance.

For the UI statistic, users provide an indicator vector ('featureIndex'), indicating a feature such as short 3′UTR isoform (SU), long 3′UTR isoform (LU), or IPA isoform for each gene. The UI statistic per gene is computed as the difference in the fraction of relative expression of this isoform in the gene across the two sets of cells. This characterizes the difference across sets of cells for a single feature.

The WUI statistic generalizes the UI statistic for genes with several isoforms by using a weighted mean of expression ratios across isoforms. scUTRboot supports the use of arbitrary weights. Throughout this work, we use a particular form we call the 'Weighted UTR Index', where the weights correspond to the positional rank of the CS from 5′ to 3′ scaled to the unit interval. That is, the shortest and longest isoforms are assigned weights of 0 and 1. An example of this is shown in Fig. S5a. Concretely, a two-isoform gene will have weights {0,1}, a three-isoform {0, 1/2, 1}, a four-isoform {0, 1/3, 2/3, 1}, and so forth.

Alternatively, users interested in statistical tests of average 3′UTR length could input the length of each isoform as the weight (not reported).

The WD statistic per gene is computed as half the total difference in all isoform expression ratios in the gene across the two sets of cells. When a gene has exactly two isoforms, the UI and WD statistics are identical in magnitude. For genes with several isoforms, the WD statistic incorporates changes in any isoform.

For each of these modes, p-values per gene are estimated using bootstrap resampling under the null hypothesis that the two sets of cells were sampled from identically distributed populations. Specifically, the union of the two sets of cells is used to sample with replacement sets of cells of the same size as the original samples. For each bootstrap sample, the statistic of interest is computed per gene and the p-value is estimated as the fraction of bootstrap statistics as extreme or greater than the observed statistic, with a pseudocount of 1 included to provide a conservative upper bound for rare events. All tests are two-sided.

scUTRboot includes a 'minCellsPerGene' option to exclude genes that are not sufficiently coexpressed in the samples to compare with

confidence. When this is set, bootstrap samples that do not satisfy this minimum are discarded, and the p-value will only be computed from the retained samples. The number of retained bootstraps samples used to estimate the p-value is included in the test results.

**Bootstrap mean TPM and WUI estimates.** For each cell type, 2000 bootstrap samples were generated by resampling with replacement from the pool of all cells with that cell type. For each bootstrap sample, two statistics were computed: a TPM value and WUI. The TPM value was computed by averaging the TPM value from across all cells in the sample, by gene.

The WUI value was computed by first summing the transcript counts across cells to pseudobulk and then computing the WUI per gene. Percentile statistics were then calculated for these values across the bootstrap samples to determine the confidence interval on the mean TPMs and mean WUIs.

**Pairwise two-sample bootstrap tests on the differentiation trajectory from HSC to Ery.** scUTRboot was used to perform two-sample WUI tests for all non-IPA isoforms detected in at least one cell type in the differentiation trajectory[46,47]. Tests were performed on all pairs of cell types (8 cell types, 28 unique pairs) using 10,000 bootstrap samples on all co-expressed genes (minimum 50 cells expressing each gene) and corrected for multiple testing using Benjamini-Hochberg procedure. Genes were classified as significant if |dWUI| > 0.10 and q-value < 0.05.

**Comparing differential gene expression with differential 3′UTR isoform length.** Differential gene expression was performed on pairs of cell types following Amezquita et al.,[70]. In brief, gene-level UMI counts were log-normalized using size factors and a pseudocount of 1. Differential expression was tested with a Welch $t$-test[73]. All p-values were corrected using the Benjamini-Hochberg procedure and genes were classified as significant if fold-changes exceeded 1.5 in either direction and q-value < 0.05.

To identify genes with differential 3′UTR isoform expression, two-sample WUI tests were performed on all cell type pairs (Fig. 3g) using scUTRboot on size-factor normalized UMI counts. All p-values were corrected using the Benjamini-Hochberg procedure and genes were classified as significant if |dWUI| > 0.10 and q-value < 0.05.

To test if gene expression and 3′UTR isoform length are independent, for each comparison, all coexpressed multi-UTR genes were classified as either non-significant, DGE only, DUL only, or both. A Chi-Square test for independence was performed on the resulting tabulation.

For each cell type pair, the fraction of genes where lower-abundance 3′UTR isoforms had larger fold-changes was computed. For each significant gene and cell type pair, the isoform with the largest absolute log-fold-change was identified in the cell type of lower expression. Then, we tested whether the isoform had lower abundance than the alternative isoform in that cell type. Only two-UTR genes were considered.

**Application of scUTRquant to a scRNA-seq dataset with 2134 perturbations**

**Processing of the Perturb-seq data set.** BAM files for the K562 6-day and RPE1 7-day essential gene Perturb-seq experiments[58] (SRA:SRR19653800-SRR19653847; SRA:SRR19653359-SRR19653414) were processed with scUTRquant using 'utrome_hg38_v1' index configured for 10x Chromium 3′ end v3. Cell annotations were extracted from the deposited H5AD objects ("K562_essential_raw_singlecell_01.h5ad"; "rpe1 _raw_singlecell_01.h5ad") and provided to scUTRquant, which transferred the perturbation annotations required for downstream analysis (https://doi.org/10.25452/figshare.plus.20029387.v1). Cells lacking a published annotation were omitted

from further analysis. Cells were summarized to pseudobulk by aggregating counts from cells with identical perturbations.

For the K562 data, we identified all isoforms in terminal exons with a mean of at least 10% of the gene expression within the 97 non-targeting perturbations, and classified genes with at least two such isoforms as multi-UTR genes in this cell line. This procedure yielded 4780 multi-UTR genes comprised of 11,129 isoforms in active use. The two highest expressed isoforms in the terminal exon of each gene were designated short UTR (SU) and long UTR (LU) with respect to their 5′ −3′ order. TPM values were calculated for all genes and all SU and LU isoforms. WUI values were computed for each multi-UTR gene in each perturbation. The rate of IPA isoform expression was computed for all genes with expressed IPA isoforms for each perturbation.

For the RPE1 data, we used the isoforms in terminals exons identified from the K562 data to compute WUI values to maintain consistency of isoform weightings when comparing across datasets. Except for the "Cluster validation with RPE1 perturbations" section, all subsequent sections pertain only to the K562 data.

The code for processing and analysis of the Perturb-seq data set is available at https://github.com/Mayrlab/gwps-sq (https://doi.org/10.5281/zenodo.10895730).

**Calculation of average dWUI and dIPA values.** To calculate the global difference in 3′UTR isoform expression between each perturbation and the 97 samples containing non-targeting guide RNAs, the average difference in WUI (dWUI) was calculated. To do so, all multi-UTR genes with a mean expression of > 5 TPM and mean WUI-values between 0.1 and 0.9 in the samples containing the 97 non-targeting guide RNAs were analyzed and the difference in the mean WUI values between the perturbation and the control samples was calculated. Similarly, a difference in IPA (dIPA) values was calculated from IPA genes with a mean expression of > 5 TPM and mean IPA-values between 0.1 and 0.9 in cells receiving the non-targeting sgRNAs cells. For each perturbation, all average dWUI and dIPA values are reported in Supplementary Data 6. These statistics average across all expressed multi-UTR or IPA genes for each perturbation. As a global average, it is most sensitive to unidirectional changes in WUI and IPA. Changes in opposite directions will cancel out.

**Clustering.** For each multi-UTR gene, we computed a baseline mean WUI value using the 97 non-targeting perturbations, weighted by the number of cells in each perturbation. We excluded genes that had more than 20% of cells non-detecting or a mean gene TPM lower than 20 in the non-targeting perturbations, leaving 1775 genes. For all perturbations with at least 30 cells ($N = 2077$), a dWUI was computed as a deviation from the baseline mean WUI. Missing WUI values in a target gene/perturbation pair were imputed as dWUI=0 (identical to baseline). A z-scaled dWUI (zdWUI) was computed by scaling the variance in dWUI across all targets to 1 without centering since the center is already determined by the non-targeting perturbations. Three rounds of clustering were then performed using these zdWUI values.

In the first round of clustering, we identified and removed sets of perturbations that showed no pattern of regulation. We first reduced the space from 1775 response genes to 30 principal components. Then, clustering was performed on the perturbations using walktrap community detection on a k = 5 nearest neighbors graph. We examined the heatmaps of zdWUI of each perturbation cluster and observed that the two largest clusters had no visible patterns. Therefore, we removed all the 1241 perturbations in these clusters from further analysis.

For the second round of clustering, we aimed to identify and remove sets of genes that showed no pattern of regulation. Dimensionality reduction and clustering were performed similarly to before, but now on the perturbation space and with k = 4. Three large clusters of genes showed no visual pattern of regulation, and their 892 genes

were removed. The third round clustered the reduced set of 836 perturbations on the reduced set of 883 target genes. This final procedure used k = 3 and identified 18 perturbation clusters and 17 target gene clusters.

**Gene ontology (GO) analysis of clustered perturbations.** The g:Profiler web tool (version e109_eg56_p17_773ec798) was used to identify enriched and depleted GO terms among the 836 gene perturbations that are cluster members relative to the background of all 2057 essential genes surveyed in the K562 dataset.

**Identification of protein complexes and pathways within perturbation clusters.** To identify protein complexes or pathways within each of the 18 perturbation clusters, we generated a protein-protein interaction network based on the genes assigned to each perturbation cluster. Data from the String database v11.5[74] were accessed through the StringApp v2.0.1 and only high-confidence physical interactions (confidence score > 0.8) were included for the network creation. Figure 4b was generated using Cytoscape v3.9.1, where average dWUI values of each perturbation were mapped to node colors and nodes were grouped by perturbation cluster. The dWUI values for each gene in each perturbation cluster are reported in Supplementary Data 7.

**Differential gene expression and differential 3′UTR length within each perturbation cluster.** For dWUI testing, we required either the set of non-targeting perturbations or the perturbations in a given cluster to have a mean TPM > 5. We then performed a two-sided Mann-Whitney test between the WUI values for each gene comparing non-targeting and targeting perturbations. P-values were corrected for multiple testing using Benjamini-Hochberg procedure. Genes were considered significant if q-value < 0.05. Differential gene expression testing was similarly performed and results for both differential gene expression and 3′UTR length are reported in Supplementary Data 8. Isoform switching analysis was performed as described in the cell type analysis section but using the sets of significant DUL genes in each perturbation cluster.

**Cluster validation with RPE1 perturbations.** To confirm the consistency of the clusters across cell types, the cluster-average zdWUIs from the K562 data were correlated with the zdWUIs of corresponding perturbations from the RPE1 data. For RPE1 perturbations, any perturbations with fewer than 30 cells were excluded, which retained 112 non-targeting perturbations and 646 perturbations that replicated a clustered perturbation in the K562 data. The 112 non-targeting perturbations were used similarly to those in the K562 data to compute dWUIs and zdWUIs for all the RPE1 data (see section on Clustering). For each cluster, the zdWUIs for response genes used in clustering that also had statistically significant dWUI tests were used to compute Pearson correlations between the cluster-average zdWUIs from K562 and zdWUIs of each replicating perturbation from RPE1. Pearson correlations were also computed with the zdWUIs of the non-targeting perturbations, providing a control distribution of random correlations for each cluster. For each cluster, a one-sided Mann-Whitney was used to test for a difference between the control correlations and the correlations of replicating perturbations for that cluster, with the alternative hypothesis of greater correlation in replicating perturbations. P-values were corrected for multiple testing using the Benjamini-Hochberg procedure.

**Overlap of genes between perturbation clusters.** To find APA regulators that target similar genes, we determined the extent of overlap within their target genes. Gene overlaps between perturbation clusters were analyzed with the GeneOverlap R package v1.32.0[75] and -log10-transformed p-values were reported (Fisher's exact test).

**Classification of shortening or lengthening by isoform-specific regulation in each perturbation cluster.** For the genes with significant differential 3′UTR expression in each cluster (Fig. 5a), we analyzed isoform-specific regulation patterns. The analysis was limited to genes that had two 3′UTR isoforms.

First, we estimated isoform expression levels within a perturbation cluster from gene expression and WUI values with:

$$\text{TPM}_{\text{SU\_Cluster}} = \text{mean}(\text{TPM}_{\text{Cluster}}) * (1 - \text{mean}(\text{WUI}_{\text{Cluster}})) \quad (1)$$

$$\text{TPM}_{\text{LU\_Cluster}} = \text{mean}(\text{TPM}_{\text{Cluster}}) * \text{mean}(\text{WUI}_{\text{Cluster}}) \quad (2)$$

Next, cluster-specific absolute TPM changes (dTPM) relative to the control conditions were calculated for each isoform:
e.g.

$$\text{dTPM}_{\text{SU\_Cluster}} = \text{TPM}_{\text{SU\_Cluster}} - \text{TPM}_{\text{SU\_Control}} \quad (3)$$

For compensatory (or balanced) 3′UTR isoform regulation, we require that at least half of the expression gained by one isoform is lost by the other 3′UTR isoform in the gene, or vice versa:
Compensatory regulation, if:

$$-0.5 \geq \frac{\text{dTPM}_{\text{LU}_{\text{Cluster}}}}{\text{dTPM}_{\text{SU}_{\text{Cluster}}}} \geq -2 \quad (4)$$

Conversely, genes for which this criterium was not met were categorized as having predominantly isoform-specific regulation, which was assigned to the isoform with the higher TPM expression difference (in absolute TPM values).

To examine the relationship between compensation and the proportion of nuclear proteins (Fig. S7b), we used the subcellular location information from the Human Protein Atlas database v22.0[76]. Proteins were classified as having nuclear localization if they had relevant terms in either the "main" or "approved" location category. The fraction of genes with such terms was calculated and plotted against the fraction of balanced DUL events in each cluster.

**Identification of gene features that correlate with dWUI in each perturbation cluster.** To investigate the relationship of gene features with 3′UTR isoform changes in each perturbation cluster we performed Pearson correlation. For all multi-UTR genes with a TPM > 5 in the non-targeting control condition, the mean dWUI value of each gene was correlated with their value for the respective gene feature. This analysis was performed for each perturbation cluster. For each feature, multiple testing correction was performed using Benjamini–Hochberg procedure and correlations with a q-value < 0.05 were considered significant.

Supplementary Data 9 lists all correlation results, along with a comprehensive list of external data sources that were used for the analysis. When identifying 3′UTR lengths and sequences, we used the coordinate of the stop codon belonging to the longest annotated coding sequence (CDS) of the gene (GENCODE v39). The same stop codon was assigned to all last exon 3′UTR isoforms of the gene and was also used for prediction of stop site readthrough[77]. Similarly, CDS and 5′UTR length, as well as other features related to their sequence, were determined based on mRNA annotations from the transcripts with the longest CDS. Features characterizing 3′UTR sequences, such as AU-rich elements, GC content and m6A modifications were assigned to both short and long 3′UTRs when occurring in the common region. For correlations related to splicing activity, a previously published dataset[78] was used to identify exon junctions with the lowest predicted splice site score for the 5′ and 3′ splice site in each gene. For analyzing correlations with m6A methylation marks, previously identified m6A sites from a HeLa dataset[79] (GEO:GSE211303) were mapped to coding regions and 3′UTRs of all multi-UTR genes expressed in K562 cells. Then, m6A scores were calculated as the sum of methylation levels at all sites within a region. For analyzing correlations with codon optimality, the fraction of non-optimal codons in each CDS was computed, where non-optimality was assigned according having a negative codon stabilization coefficient (CSC) score in a K562 SLAM-seq experiment[80]. The R package 'codonopt' was created to compute codon optimality with these CSC scores, and is available at https://github.com/mfansler/codonopt (https://doi.org/10.5281/zenodo.10845963).

**Isoform-specific mRNA half-life analysis.** A K562 SLAM-seq experiment was used for mRNA stability analysis[80] (GEO:GSE126522). FASTQ files were processed using the SLAM-DUNK pipeline[81] with our MWS UTRome as reference annotation. For each isoform, the mean conversion rate per timepoint (0, 120, 240, 360 min) was computed as the weighted mean of conversion rates weighted by read counts ('ReadsCPM') across the three replicates. Mean conversion rates per isoform were modeled with first-order kinetics with the R 'glm' function, and mRNA half-life computed from the coefficients. The processing pipeline is available at https://github.com/Mayrlab/slam-k562-utrome (https://doi.org/10.5281/zenodo.10887802).

### Quantification and statistical analysis
Statistical tests used throughout this study are indicated in the Methods section. Unless otherwise indicated, statistical significance testing was two-sided, and an FDR of 5% was used after the Benjamini-Hochberg adjustment. Figure legends document the statistics indicated by bars, error bars, and box-whiskers.

### Reporting summary
Further information on research design is available in the Nature Portfolio Reporting Summary linked to this article.

## Data availability
This paper analyzes existing, publicly available data. The accession numbers for the datasets are listed below and in the corresponding Methods sections discussing their processing. The Mouse Cell Atlas v1.1[36] data used in this study are available in the GEO database under accession code GSE108097. The Human Cell Landscape[37] data used in this study are available in the GEO database under accession code GSE134355. The bulk 3′-seq FACS-sorted HSCs data[45] used in this study are available in the ArrayExpress database under the accession code E-MTAB-7391. The scRNA-seq mouse HSPC data[46,47] used in this study are available in the GEO database under the accession code GSE107727. The mouse ESC scRNA-seq data[50] used in this study are available in the GEO database under the accession codes GSM3629847, GSM3629848, and GSM4694997. The mouse brain data[56] used in this study are available in the GEO database under the accession code GSE129788. The Tabula Muris data[55] used in this study are available in the GEO database under the accession code GSE109774. The raw Tabula Sapiens data[54] used in this study is on AWS under restricted access due to data privacy restrictions; access can be requested at https://tabula-sapiens-portal.ds.czbiohub.org/whereisthedata. The K562 6-day essential gene Perturb-seq experiments[58] used in this study are available in the SRA database under accession codes SRR19653800-SRR19653847 [https://www.ncbi.nlm.nih.gov/Traces/study/?acc=SAMN28561243]. The RPE1 RPE1 7-day essential gene Perturb-seq experiments[58] used in this study are available in the SRA database under accession codes SRR19653359-SRR19653414 [https://www.ncbi.nlm.nih.gov/Traces/study/?acc=SAMN28561244]. The cell annotations for the Perturb-seq experiments[58] used in this study are available on figshare (https://doi.org/10.25452/figshare.plus.20029387.v1). The m6A sites mapped by eTAM seq in HeLa data[79] used in this study are available in the GEO database under accession code GSE211303. The K562 SLAM-seq data[80] used in this study are available in the GEO

database under accession code GSE126522. The codon stability coefficients data used in this study are from Fig. 1-Source Data 2[80] available at https://doi.org/10.7554/eLife.45396.006. The PolyASite 2.0 databases[35] for human and mouse used in this study are available at https://polyasite.unibas.ch/atlas. The PolyA_DB v3.2 database[34] for human and mouse used in this study are available at https://exon.apps. wistar.org/polya_db/v3/misc/download.php. The protein subcellular localization data used in this study are available in the Human Protein Atlas database (v22.0)[76] at https://www.proteinatlas.org. The STRING Database v11.5 data[74] used in this study are available at https://version-11-5.string-db.org/. Processed data generated in this study are available in the following locations: The GTF and BED files for the MWS CS annotation in human and mouse data generated in this study are available on figshare (https://doi.org/10.6084/m9.figshare.23549526). The SingleCellExperiment and SummarizedExperiment objects generated in this study by scUTRquant and post-processing pipelines are available on figshare (https://doi.org/10.6084/m9.figshare.25513528). The intermediate data objects used to generate figures are available on figshare (https://doi.org/10.6084/m9.figshare.25529632). The MWS CS annotations for human generated in this study are provided in Supplemental Data 1. The MWS CS annotations for mouse generated in this study are provided in Supplemental Data 2. The human gene annotations from analyzing 355 human cell types generated in this study are provided in Supplemental Data 3. The mouse gene annotations from analyzing 119 mouse cell types generated in this study are provided in Supplemental Data 4. The independence and correlation test results between significant DGE and DUL generated in this study are provided in Supplemental Data 5. The average dWUI and dIPA for K562 6-day essential perturbations generated in this study are provided in Supplemental Data 6. The APA regulator clusters and GO term analysis generated in this study are provided in Supplemental Data 7. The DGE and DUL significance testing results for APA regulator clusters generated in this study are provided in Supplemental Data 8. The correlation test results among DUL changes and genomic features generated in this study are provided in Supplemental Data 9.

## Code availability

All original code has been made publicly available on Github as of the date of publication. The repositories for processing pipelines are listed below and in the corresponding Methods sections: The code to generate the human MWS annotation is deposited at https://github.com/Mayrlab/hcl-utrome (https://doi.org/10.5281/zenodo.8118411). The code to generate the mouse MWS annotation is deposited at https://github.com/Mayrlab/mca-utrome (https://doi.org/10.5281/zenodo.8118415). The code to process bulk HSPC data with the mouse MWS annotation is deposited at https://github.com/Mayrlab/sommerkamp20 (https://doi.org/10.5281/zenodo.10892209). The code to process SLAM-seq data and compute isoform-specific mRNA half-lives is deposited at https://github.com/Mayrlab/slam-k562-utrome (https://doi.org/10.5281/zenodo.10887801). The code to generate custom truncated transcriptomes for the scUTRquant pipeline is deposited at https://github.com/Mayrlab/txcutr-db (https://doi.org/10.5281/zenodo.8118404). The repositories for original software intended for reuse are listed below and in the corresponding Methods sections: The scUTRquant pipeline is deposited at https://github.com/Mayrlab/scUTRquant (https://doi.org/10.5281/zenodo.8118393). The source code for the customized version of kallisto is deposited at https://github.com/mfansler/kallisto/releases/tag/v0.46.2sq (https://doi.org/10.5281/zenodo.10902020). The scUTRboot R package is deposited at https://github.com/mfansler/scutrboot (https://doi.org/10.5281/zenodo.8057843). The txcutr Bioconductor package is deposited at https://bioconductor.org/packages/txcutr (https://doi.org/10.18129/B9.bioc.txcutr). The codonopt R package for computing codon optimality is available at https://github.com/mfansler/codonopt (https://doi.org/10.5281/zenodo.10845962). The repositories for analyses and figures

presented in the manuscript are listed below: The code to characterize the human MWS annotation is deposited at https://github.com/Mayrlab/hcl-analysis (https://doi.org/10.5281/zenodo.10892181). The code to characterize the mouse MWS annotation is deposited at https://github.com/Mayrlab/mca-analysis (https://doi.org/10.5281/zenodo.10892185). The code to characterize peak widths in Tabula Muris data is deposited at https://github.com/Mayrlab/tmuris-peaks (https://doi.org/10.5281/zenodo.10895190). The code to characterize kallisto resolution for overlapping transcripts is deposited at https://github.com/Mayrlab/kallisto-overlap (https://doi.org/10.5281/zenodo.10895237). The code to characterize isoform expression across the Tabula Sapiens dataset is deposited at https://github.com/Mayrlab/atlas-hs (https://doi.org/10.5281/zenodo.10895336). The code to characterize isoform expression across mouse datasets is deposited at https://github.com/Mayrlab/atlas-mm (https://doi.org/10.5281/zenodo.10895351). The code to process and analyze the Perturb-seq data set is deposited at https://github.com/Mayrlab/gwps-sq (https://doi.org/10.5281/zenodo.10895730). Additional code to generate analyses and figures is deposited at https://github.com/Mayrlab/scUTRquant-figures (https://doi.org/10.5281/zenodo.10910013).

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

## Acknowledgements

This work was funded by NIH training grant T32GM083937 to M.M.F., the NIH Director's Pioneer Award (DP1-GM123454), the R35GM144046 NIH grant, a grant from the Pershing Square Foundation, William Ackman, and Neri Oxman, and the MSK Core Grant (P30 CA008748) to C.M. We thank Quaid Morris and all members of the Mayr lab for helpful discussions. We also thank Andrew Grimson, Lucy Skrabanek, Steve Lianoglou, and Julia Simundza for useful comments on the manuscript.

## Author contributions

M.M.F. and C.M. conceived the project. M.M.F. developed and implemented all new computational software and processing pipelines, generated the MWS annotation, processed all datasets, and identified the clusters from Perturb-seq data. S.M. analyzed the Perturb-seq clusters and characterized the associated gene features. All authors designed the experiments and wrote the manuscript.

## Competing interests

The authors declare no competing interests.
