## [Peer Review File · Nature Communications]

Quantifying 3'UTR length from scRNA-seq data reveals changes independent of gene expressionREVIEWER COMMENTS

Reviewer #1 (Remarks to the Author):

In the manuscript by Fansler et.al, the authors presented a systematic analysis of 3'UTR/PAS profile based on single cell RNA-seq data. First, they took advantage of microwell-seq data of 206 human and mouse cell types and annotated over 53,000 and 45,000 PAS sites in human and mouse, respectively. Based on the no. of cell type where these sites are used, the PAS sites could be characterized into major and minor ones. Intriguingly, most of genes have used only one or two major sites, which is in contrast to current PAS annotation which suggested that most genes used >5 PAS sites. Then, the authors developed a computational pipeline, scUTR quant for 3'UTR isoform quantification based on scRNA-seq data and validated its performance by comparing their results with those from bulk 3'seq data. Finally, by applying their tool to scRNA-seq data from cell differentiation, different cell types as well as genetic perturbation, they discovered 1) changes in UTR isoform usage is as widely as those in gene expression, but the two hit different gene sets. This resource expanded the commonly used Genecode annotation by 40%. plate-based high-throughput method to quantify ribosome-associated RNAs. The so-termed riboPLATE-seq is an extension of PLATE-seq, a protocol previously developed in the same lab. The authors then combined the two methods to estimate translation efficiency in cells treated with different kinase inhibitors.

Overall, the method described in the manuscript represented a rather limited technical advance. Moreover, the authors did not sufficiently evaluate the performance. Finally, the analysis of their data derived from different drug treatments is superficial and does not provide novel mechanistic understanding of translational regulation.

More specific points are listed below,

1. Comparing to Gencode annotation, what is the distribution of usage of novel sites versus common sites?
2. It would be interesting if the authors provide a figure/table summarizing the distribution of genes using more than one PAS site ($\geq 10\%$ usage) simultaneously in the different no. of cell types. This can reflect the APA regulation within the same cell type.
3. When GE and UTR isoform were compared between different samples, the two were often regulated in different set of genes. Have the authors checked whether GE was more likely changed for single UTR genes, in particular for the perturbation of core CPA machinery?
4. Since scUTRquant has limitation in distinguishing PAS sites in vicinity, how will this affect their conclusion, including the overlap of change in between GE and 3'UTR isoform, the clustering of gene perturbation etc.?
5. In all the perturbation experiments, are there any APA switch events, i.e. the switch of dominant 3'UTR isoforms? If so, what is the distribution of their proportions?
6. Change in RNA abundance and isoform usage could be due to either their biogenesis or decay. They should perform at least one perturbation for each of the 18 clusters and use bulk 3'seq to validate their findings, and then use 4sU labeling or transcription inhibition to distinguish the effects between processing and differential decay.
7. To a certain extent, the usage of different PAS sites is a competitive process, similar as alternative splicing. Why in figure 5d, most of 3'UTR isoform changes do not belong to the middle panel?

Reviewer #2 (Remarks to the Author):

Summary: Fansler et al constructed a comprehensive and cell type-resolved cleavage site (CS) annotation in human and mouse by mapping reads overlapping 3' cleavage sites from >200 Microwell-seq (MWS) data (collected from the Human Cell Landscape and Mouse Cell Atlas projects). The authors performed a number of post-processing steps to reduce artifacts (e.g. internal priming) and compared their measurements to orthogonal tools (e.g. APARENT2) and datasets (e.g. conservation scores, overlap with GENCODE-, PolyADB- and PolyASite annotations, etc.) for quality control. The authors demonstrated the utility of their new annotation for 3' isoform quantification in scRNA-seq data, with applications ranging from erythroid differentiation to analyzing scPerturb-seq data. By combining their 3' isoform quantification with network- and pathway analyses, the authors make pathway-specific observations about overall trends in global UTR shortening or lengthening upon perturbation.

Impact: Overall I found this to be a nicely executed study that resulted in a much-needed consensus annotation/quantification of cleavage sites and their usage in diverse human cell types. The axis of cell type-specificity has been largely absent from earlier pA site atlas efforts so I expect this will be of wide use for the research community. The authors perform a lot of QC to verify their annotation. Additionally, the authors spend a good deal of effort using their annotation to analyze a large perturb-seq dataset and draw interesting conclusions about general shortening trends across different regulatory pathways, and quantify the (surprising, but increasingly more accepted) observation that gene expression and shortening is less coupled than what was earlier believed.

I have a few comments and suggestions regarding some of the analyses. After these have been addressed, I recommend accepting the manuscript for publication.

Major comments: I have no major comments / criticisms.

Minor comments:

Minor #1: In the paragraph starting on Line 112 (and associated Fig. 1g), one comparison that I feel is missing is that between "CS usage" (as authors define it: frequency of being used at all across cell types) and isoform abundance proportion (i.e. how much the CS is used in comparison to competing proximal or distal CS in the same gene and cell type). I suggest adding a (simple) comparison of some sort (e.g. scatter plot or histograms) comparing these metrics. Otherwise it's not clear to me whether the cell type-common cleavage sites are lowly-used/weak signals or strong/canonical signals (though I'm guessing it's the latter).

Minor #2: It might be prudent to add some hypothesis or reason for why the replicate concordance is so much better in scRNA-seq data than in bulk 3' sequencing data, given the same annotation for quantification. Is it just a function of read depth?

#Minor 3: Regarding the paragraph starting at Line 166 (and associated Fig. 3a and onward) : Does UTRome (used for scUTRquant) include intronic pA annotations, or only 3' UTR CS events? It was not immediately clear to me (though in the next paragraph the authors mention IPA isoforms so I assume yes). It may be worth clarifying this in the paragraph at Line 166.

#Minor 4: Suppl Fig S5D: Is it unexpected that "Distal PAS Score" (which is APARENT2 score?) promotes overall differential shortening upon most pathway perturbations? I would have naively guessed that a stronger distal PAS in general would be associated with lengthening.

Reviewer #3 (Remarks to the Author):

In this paper by Fansler et al from the Mayr lab, the authors have created a pipeline (scUTRquant) for the identification of human and mouse cleavage sites (CS) from very large amounts of single cell RNA-seq data (>100 datasets from each of mouse and human), resulting in the identification of ~40% more CS than are present in GENCODE. They conclude that most of the CS not present in GENCODE are used in only very few cell types and that most genes have only 1 or 2 CS that are used in the vast majority of cell types. They then examine single cell perturb-seq data published last year by the Weissman lab, focusing on CRISPRi-generated knock-downs of 2134 essential genes in order to compare effects on gene expression and CS choice. One central conclusion is that effects on CS choice are as common as those on gene expression. Another is that effects on gene expression and CS choice are mostly independent of each other, indicating that CS choice mostly affects processes other than gene expression (e.g. mRNA localization). Finally, by clustering the knockdowns that affect CS choice for similar gene groups, they identify 18 groups of genes that differ in the groups of genes whose CS choice they affect. They also conclude that effects on CS choice mostly affect the proximal CS, even though CFI, which has the biggest effect, affects distal sites. There is much that is praiseworthy about this paper, especially its completeness for the phenomena that are described. On the other hand, the impact of the paper will be somewhat limited, both because the independence of gene expression and CS choice has already been described in multiple papers based on less complete data, and because the analysis of the perturb-seq data is limited by its lack of mechanistic insight.

Major points:

1. The authors point out that the CS they identify that are not present in GENCODE are generally found in PolyA-DB v3.2 and PolyASite 2.0, which means that most of the "new" CS they identify are not really new, only not present in GENCODE. Therefore, what is really new is the observation that most of the CS in the existing databases are rarely used (i.e. used in only a very few cell types). This needs to be made more clear.
2. As the authors themselves point out, the independence of gene expression effects from CS effects, which is actually the title of the paper, has already been described for several specific cell types by papers from multiple labs, including the Mayr group (e.g. references 2, 23, and 24 from 2013 and 2014). This manuscript essentially generalizes that finding to comparisons of essentially any cell types.
3. The 18 gene groups that affect CS choice encompass a total of 836 genes out of the 2134 genes whose knockdowns were examined. That amounts to ~40% of the genes. On the one hand, that seems remarkable, because it is hard to believe that so many proteins could directly influence CS choice. However, these are all essential proteins, and they either affect cell cycle or essentially every aspect of the gene expression pathway (transcription [2 groups], chromatin, splicing [2 groups], CPA, mRNA export, translation [3 groups]). It is obvious that most of these effects must be indirect, perhaps very indirect. In fact, it is possible that none of the effects are direct except those of the cleavage factors themselves. Because of this lack of mechanistic insight, all we learn is that seriously messing up the cell almost always affects cleavage.
4. Another downside of the analysis of the perturb-seq data is that it is never clear whether it is actually CS choice that is being affected. No matter which one of the 18 groups is being considered (with the exception of the cleavage factors themselves), one cannot tell whether the effect is on differential stability of the mRNA isoforms or else on CS choice.
5. It is not obvious that the 18 gene groups that affect CS choice are really distinct from one another. For example, the gene groups affected by splicing group I, splicing group II, and transcription group I all look similar. Also, the effects of the tRNA synthesis group, the DNA replication and repair group, transcription group II, CPSF, and the nuclear exosome look quite similar. Finally, the three translation groups look similar.

Minor point: there are some issues with the reference list. The first citation for a review on CPA only mentions one from the Mayr lab, and there are many other good ones. This is followed by the citation only of a Mayr lab paper showing how many genes use APA, and again there are others. Finally, there are references in the list that are missing various things (e.g. references 27, 28, 33).

REVIEWER COMMENTS

Reviewer #1 (Remarks to the Author):

In the manuscript by Fansler et.al, the authors presented a systematic analysis of 3'UTR/PAS profile based on single cell RNA-seq data. First, they took advantage of microwell-seq data of 206 human and mouse cell types and annotated over 53,000 and 45,000 PAS sites in human and mouse, respectively. Based on the no. of cell type where these sites are used, the PAS sites could be characterized into major and minor ones. Intriguingly, most of genes have used only one or two major sites, which is in contrast to current PAS annotation which suggested that most genes used >5 PAS sites. Then, the authors developed a computational pipeline, scUTR quant for 3'UTR isoform quantification based on scRNA-seq data and validated its performance by comparing their results with those from bulk 3'seq data. Finally, by applying their tool to scRNA-seq data from cell differentiation, different cell types as well as genetic perturbation, they discovered 1) changes in UTR isoform usage is as widely as those in gene expression, but the two hit different gene sets. This resource expanded the commonly used Genecode annotation by 40%.

We thank the reviewer for spending time with our manuscript and for the constructive comments.

More specific points are listed below,

1. Comparing to Gencode annotation, what is the distribution of usage of novel sites versus common sites?

Response: We followed the reviewer's suggestion and assessed CS usage rates as a function of read count fractions (TPM of isoform/total TPM of gene) across many cell types. For all CS in multi-UTR genes, we plotted the median expression ratio of the 3'UTR isoforms for 234 human and 82 mouse cell types with more than 200 cells each. CS isoforms were grouped into "common", "MWS-only" (microwell-seq only) or "GENCODE-only", depending on the annotation category of the associated CS. In response to Reviewer #2's first question, we also performed a similar analysis where isoforms were grouped into major and minor CS, respectively.

*The results of these new analyses are now presented in **new Fig. S3k and S3l** of the manuscript. They show that transcripts produced from usage of "common" and "major" sites contribute the highest fractions to total gene expression, further establishing these sites as the predominantly used CS. Novel sites are used less, but a quarter of the novel sites is have expression levels higher than 30% (human) or higher than 40% (mouse) with respect to the total expression of the gene. In contrast, CS that are listed in the GENCODE annotation, but were not validated through MWS are expressed at low levels. These isoforms only had a median expression level amounting to 6% and 4% of total gene counts in human and mouse across the cell types where the isoform was detectable. Similarly, 3'UTR isoforms from minor CS also showed low expression.*

Please note, that while our CS atlas created from MWS data has high resolution, the quantification of isoform expression from typical single cell 10x genomics data has more limited resolution and often requires merging of read counts from closely-spaced CS (see Figure S2b).

Therefore, for these analyses, we focused on CS that do not undergo merging and where all reads can be unambiguously assigned (about one third of all CS in multi-UTR genes).

Overall, these results closely mirror other CS quality metrics that we had previously analyzed (see Fig. 1c, 1d, 1e and Fig. S1a, S1b, S1c).

2. It would be interesting if the authors provide a figure/table summarizing the distribution of genes using more than one PAS site ($\geq 10\%$ usage) simultaneously in the different no. of cell types. This can reflect the APA regulation within the same cell type.

Response: To address this suggestion, we analyzed the fraction of multi-UTR genes expressing either one or more 3'UTR isoforms across 234 human and 82 mouse cell types. These data are now included in the manuscript **in new Fig. S3g-j**. The analysis reveals that some multi-UTR genes (~20%) functionally resemble single-UTR genes in that they mostly express a single dominant isoform. We believe that these may be linked to minor CS that consistently produce low expression levels (see **Fig S3k, S3l**). However, the majority of multi-UTR genes simultaneously express two or more isoforms in all cell types.

It is not entirely clear to us what is meant by "This can reflect the APA regulation within the same cell type". It could be that the reviewer wants to see analysis of APA at the 'real' single cell level. However, the coverage in the data is not high enough to perform APA analysis at the level of single cells, which is the reason why we pool all the cells from one cell type and perform APA analysis across cell types.

3. When GE and UTR isoform were compared between different samples, the two were often regulated in different set of genes. Have the authors checked whether GE was more likely changed for single UTR genes, in particular for the perturbation of core CPA machinery?

Response: The reviewer put forward an interesting question. Our study showed that gene knockdowns associated with differential gene expression (DGE) were largely identical to those regulating differential 3'UTR length (DUL). It is tempting to speculate that the type of regulation is largely defined by the class of gene, and, in particular, that DGE is more common among single- than multi-UTR genes.

We investigated this hypothesis by calculating the fraction of single- and multi-UTR genes among the DGE genes for each of the 18 perturbation clusters (see **new Fig. S5j**). Surprisingly, DGE was at least as likely - or more likely - to occur in multi-UTR genes than in single UTR genes. This was also true for the perturbation clusters encoding genes that are directly involved with the cleavage machinery (e.g. cluster 1: CFlm cluster; cluster 17: CPSF, CSTF and PAF complex). Our results suggest that multi-UTR genes can change both, 3'UTR length or gene expression. As these genes rarely change both parameters at the same time, our data indicate that in one context these genes change expression, whereas in a different context, they change 3'UTR length.

4. Since scUTRquant has limitation in distinguishing PAS sites in vicinity, how will this affect their conclusion, including the overlap of change in between GE and 3'UTR isoform, the clustering of gene perturbation etc.?

Response: Due to the broad peak width and low frequency of CS-traversing reads from the 10x sequencing data, distinct isoforms produced from closely-spaced CS cannot be resolved within the framework of kallisto with an acceptable error rate (Fig. S2b). Therefore, our data processing pipeline, scUTRquant, was configured to instead provide a summed UMI count for such regions, effectively merging these CS.

The reviewer is wondering how our results might change if scUTRquant could provide full resolution in the isoform quantification. We first point out that our gene-level quantification would be unimpacted by such a change, and therefore we would expect differential gene expression testing results to be unchanged throughout the work. Hence, the primary difference would be in potentially classifying more genes as multi-UTR in the classifications reported in **Fig. S3a-f**.

We investigated whether merging of reads affects our conclusion of independence between DGE and DUL. Since we cannot distinguish reads from the 10x data directly, we decided to investigate whether genes where merging occurs have similar independence of DUL and DGE changes to those genes where reads were assigned unambiguously. For 14/17 of the comparisons, we arrive at identical conclusions whether we consider only the unambiguous genes, only the genes with merging, or the (original) combined test. The three comparisons with conclusion differences are provided in the following table.

	p-value (chi-squared test)		
	Unmerged	Merged	Combined
Neutro vs HSC	0.228	0.047	0.248
ODC vs OPC	0.002	0.676	0.029
TraEpi vs TraMes	0.114	0.081	0.028

As the majority of tests (14/17) result in identical conclusions, our results suggest that merging of isoforms does not alter our conclusions about the independence of DUL and DGE events.

Respecting the analysis of the Perturb-seq data, there could be a slight chance of obtaining more fine-grained clustering, but we expect it unlikely to have any substantial impact on the broad structure. The Perturb-seq data for any particular perturbation and any particular gene is sparse and thus extremely noisy. Whereas, the clustering analysis is coarse-grained, in the sense that it relies on common patterns of variation both across many perturbations and across many genes to establish communities. Unless there were a hidden pattern of WUI changes common to a substantial fraction of additional multi-UTR genes, we expect almost no change to the clusters that were identified. Similarly, the feature correlations are computed using the dWUI value from all genes, and thus would not be expected to substantially vary.

5. In all the perturbation experiments, are there any APA switch events, i.e. the switch of dominant 3'UTR isoforms? If so, what is the distribution of their proportions?

Response: The reviewer is interested to know to what extent each of the perturbation clusters causes strong enough DUL changes, such that the highest expressed 3'UTR isoform within a gene changes relative to the non-targeting control condition. We will call this phenomenon isoform switching.

To answer this question, we analyzed the fraction of isoform switching events among all DUL genes and within each perturbation cluster. Among multi-UTR genes for which we recorded DUL changes in any cluster, 26% exhibited isoform switching in at least one perturbation condition (**new Fig. S6a**). Switching behavior varied between perturbation clusters, ranging from 6% to 32% of significant DUL events (see **new Fig. S6a**).

However, while the Perturb-seq analysis is an excellent tool to identify novel regulators of APA, we think that the effect sizes seen in these experiments are not comparable to those in cell types. It seems that larger DUL changes occur when several regulatory modules change their activity simultaneously. In order to estimate the degree to which isoform switching occurs in the broader contexts of endogenous gene regulation, we performed further analyses of human and mouse cell type data. We found that a majority of multi-UTR genes can undergo isoform switching in at least one endogenous cell type (60% and 52% of human and mouse multi-UTR genes, respectively, **new Fig. 3b**). We think that this analysis highlights the diversity of the 3'UTR isoform expression landscape and the benefits of an analysis pipeline that can resolve these differences.

6. Change in RNA abundance and isoform usage could be due to either their biogenesis or decay. They should perform at least one perturbation for each of the 18 clusters and use bulk 3'seq to validate their findings, and then use 4sU labeling or transcription inhibition to distinguish the effects between processing and differential decay.

Response: We believe that the reviewer makes two propositions: 1. Perform validations of 3'UTR isoform expression regulators in a different system. 2. Investigate the extent to which either biogenesis or decay of mRNA isoforms is responsible for the observed differences.

To address (1), we compared the changes in 3'UTR isoform expression (as measured by a z-scaled dWUI) obtained in K562 cells in each perturbation cluster to the corresponding changes obtained in a different cell line (RPE1 cells, which are adherent epithelial cells as opposed to suspension cells derived from the myeloid lineage). As shown in the **new Fig. S5i**, the Pearson's correlation coefficients of the z-scaled dWUI values obtained from the two cell lines are all significant (one-sided Mann Whitney test), when comparing them to a random distribution of z-scaled dWUI values obtained from the non-targeting controls (**see methods, lines 794-795, 812-815 and 873-887 for details**). The strongest correlations were observed for splicing and mRNA export factors, as well as for the proteasome and cell cycle-regulated genes. We therefore conclude that the distinct patterns of altered 3'UTR isoform expression observed upon knockdown of genes assigned to the different functional clusters are largely reproducible in a different cell system.

With respect to (2), we agree with the reviewer, that it would be interesting to disentangle the effects of differential processing from decay in each perturbation cluster. However, a thorough analysis of these processes on the isoform level across a large number of perturbations would require new experimental approaches alongside sophisticated data analysis methods. We believe that such an investigation goes beyond the scope of this paper, but we hope that these challenges will be addressed in the future.

7. To a certain extent, the usage of different PAS sites is a competitive process, similar as alternative splicing. Why in figure 5d, most of 3'UTR isoform changes do not belong to the middle panel?

To our knowledge, it is unclear to what extent CS choice is truly a competitive process and the experimental evidence on this subject is very limited. The competitive-use-model predicts that a relative decrease in use of one CS is accompanied by an increase in use of other CS within the gene. As a result, the transcripts generated from the different CS are expected to show oppositely coordinated or balanced expression patterns. Our data analysis reveals that such a pattern of regulation is less frequently observed than changes of individual isoforms (Fig. 5d). We have two hypotheses regarding this finding.

*First, our dataset (as well as all other published APA datasets) quantifies steady-state transcript expression levels, which are equally determined by both biogenesis and decay. Whereas competition between CS may occur during transcript biogenesis, the expression regulation of different 3'UTR isoforms is physically uncoupled in later stages of the mRNA life cycle. It is unlikely that a mechanism leading to increased decay of one 3'UTR isoform causes the opposite outcome for the other isoform. Therefore, posttranscriptional processing steps (including export, translation, mRNA storage and degradation) are not expected to result in compensatory changes in 3'UTR isoform abundance. Indeed, we found that the fraction of DUL events with balanced transcript changes was highest for clusters containing mostly nuclear proteins (**new Fig. S6b**). This supports the hypothesis that coordinated transcript regulation is restricted to processes impacting CS choice and does usually not occur during posttranscriptional regulation.*

*Second, we find that even among the clusters involved in co-transcriptional processing, there is a high fraction of isoform-specific DUL events, including those associated with perturbations of the cleavage and polyadenylation machinery (clusters 1 and 17). This suggests that there may be further limitations to the competitive-use-model. Based on previous studies on this topic, we think that some CS may require additional positive stimuli to be used effectively. This means, for example, that strong CS often contain NUDT21 binding sites, which are predominantly observed at distal CS. When a strong CS is encountered, the majority of transcripts that arrive at this site get processed. However, if NUDT21 is lacking (cluster 1) these usually strong CS become weaker and only a small fraction of transcripts is processed at these CS. If the distance between CS is sufficiently large, this results in lower expression of the longer transcripts without affecting expression of the shorter transcripts. It seems that NUDT21 is only one example from a large range of positive APA regulators. In the revised manuscript, we added additional text (**lines 365-383**) to clarify.*

We think that it is commonly assumed that the predominant pattern for APA changes are compensatory changes. However, our large-scale analysis (Fig. 5d) showed that this is not true.

*In our opinion, this misconception might have been caused in part by the words chosen to describe the process (alternative cleavage and polyadenylation), which implies that the observed changes happen during biogenesis. Moreover, the word 'usage' is often used (including by us) to describe relative expression of 3'UTR isoforms, but it implies that the regulation happens at the biogenesis step. Therefore, to emphasize that the conclusions drawn in this manuscript are from steady-state levels of 3'UTR isoforms (which can occur through changes in biogenesis or degradation), we changed the wording from '3'UTR isoform usage' in all instances **throughout the manuscript** to '3'UTR isoform expression'.*

Taken together, independent regulation of 3'UTR isoforms is likely a common feature of gene regulation, thus, highlighting the need for appropriate pipelines for reliable and accurate 3'UTR isoform quantification.

Reviewer #2 (Remarks to the Author):

Summary: Fansler et al constructed a comprehensive and cell type-resolved cleavage site (CS) annotation in human and mouse by mapping reads overlapping 3' cleavage sites from >200 Microwell-seq (MWS) data (collected from the Human Cell Landscape and Mouse Cell Atlas projects). The authors performed a number of post-processing steps to reduce artifacts (e.g. internal priming) and compared their measurements to orthogonal tools (e.g. APARENT2) and datasets (e.g. conservation scores, overlap with GENCODE-, PolyADB- and PolyASite annotations, etc.) for quality control. The authors demonstrated the utility of their new annotation for 3' isoform quantification in scRNA-seq data, with applications ranging from erythroid differentiation to analyzing scPerturb-seq data. By combining their 3' isoform quantification with network- and pathway analyses, the authors make pathway-specific observations about overall trends in global UTR shortening or lengthening upon perturbation.

Impact: Overall I found this to be a nicely executed study that resulted in a much-needed consensus annotation/quantification of cleavage sites and their usage in diverse human cell types. The axis of cell type-specificity has been largely absent from earlier pA site atlas efforts so I expect this will be of wide use for the research community. The authors perform a lot of QC to verify their annotation. Additionally, the authors spend a good deal of effort using their annotation to analyze a large perturb-seq dataset and draw interesting conclusions about general shortening trends across different regulatory pathways, and quantify the (surprising, but increasingly more accepted) observation that gene expression and shortening is less coupled than what was earlier believed.

I have a few comments and suggestions regarding some of the analyses. After these have been addressed, I recommend accepting the manuscript for publication.

We thank the reviewer for spending time with our manuscript, for the positive assessment, and for the constructive comments.

Major comments: I have no major comments / criticisms.

Minor comments:

Minor #1: In the paragraph starting on Line 112 (and associated Fig. 1g), one comparison that I feel is missing is that between “CS usage” (as authors define it: frequency of being used at all across cell types) and isoform abundance proportion (i.e. how much the CS is used in comparison to competing proximal or distal CS in the same gene and cell type). I suggest adding a (simple) comparison of some sort (e.g. scatter plot or histograms) comparing these metrics. Otherwise it’s not clear to me whether the cell type-common cleavage sites are lowly-used/weak signals or strong/canonical signals (though I’m guessing it’s the latter).

Response: We have now added two **new figures (Fig. S3k, S3l)** that describe the median fraction of reads that originate from isoforms generated from either major or minor CS across 234 human and 82 mouse cell types. We also performed a similar analysis for isoforms grouped by CS atlas annotation (see Rev #1, question #1).

The results of our analyses demonstrate that isoforms associated with minor CS typically are expressed at low levels. The median fractions of a gene’s total expression counts assigned to minor CS were 6% and 4% in human and mouse, respectively. However, a subset of these CS may contribute significantly to higher expression levels in the cell types that they are expressed. Specifically, 14% and 11% of human and mouse minor CS isoforms generate at least one third of their gene’s total expression (**Fig. S3k, S3l**). This suggests that these minor isoforms may be more likely to have important cell-type-specific functions.

Please note that isoform quantification from 10x Genomics data often requires merging of read counts from closely-spaced CS (see Fig. S2b). Therefore, we only analyzed expression associated with those CS where all reads can be unambiguously assigned (about one third of all CS in multi-UTR genes).

Minor #2: It might be prudent to add some hypothesis or reason for why the replicate concordance is so much better in scRNA-seq data than in bulk 3’ sequencing data, given the same annotation for quantification. Is it just a function of read depth?

Response: In addition to read depth, we investigated if removal of PCR duplicates (through use of unique molecular identifiers) in scRNA-seq data is responsible for better concordance. However, although scRNA-seq data were approximately an order of magnitude deeper, downsampling of the scRNA-seq data did not negatively impact the correlations across the replicates.

Spearman ρ	bulk-bulk, N = 6¹	bulk-sc, N = 12¹	sc-sc, N = 3¹
original	0.909 (0.907, 0.915)	0.856 (0.847, 0.863)	0.987 (0.987, 0.987)
downsampled	0.908 (0.907, 0.914)	0.856 (0.848, 0.861)	0.977 (0.977, 0.977)
¹ Median (IQR)			

To test if the use of UMIs to remove PCR duplicates affects replicate correlations, we reprocessed the scRNA-seq mouse ESCs samples as bulk data by processing only the biological reads directly with kallisto. This effectively ignores all barcode and UMI information and does not filter ambient RNA or reads from low-quality cells. However, this again had no significant negative impact on the sample correlations across experimental replicates.

Spearman ρ	bulk-bulk, N = 1¹	bulk-sc, N = 4¹	sc-sc, N = 1¹
original	0.794 (0.794, 0.794)	0.644 (0.604, 0.689)	0.939 (0.939, 0.939)
as bulk	0.794 (0.794, 0.794)	0.642 (0.603, 0.686)	0.941 (0.941, 0.941)
¹ Median (IQR)			

*As none of the parameters tested had any influence on the correlations of the replicates, we speculate that standardized chemistry and automated workflows used by the single-cell sequencing platforms reduce sample variance. In contrast, when performing bulk 3' end sequencing, research labs use 'home-made' methods that may have not been sufficiently optimized. Moreover, each laboratory uses its own library preparation protocol and own reagents, which may contribute to the larger variation. We have now included a statement in the text (**lines 166-169**), where we discuss this topic.*

#Minor 3: Regarding the paragraph starting at Line 166 (and associated Fig. 3a and onward) : Does UTRome (used for scUTRquant) include intronic pA annotations, or only 3' UTR CS events? It was not immediately clear to me (though in the next paragraph the authors mention IPA isoforms so I assume yes). It may be worth clarifying this in the paragraph at Line 166.

*Response: The UTRome includes annotation for last exon as well as intronic 3' ends of all protein-coding genes. In the annotation, sites are marked with regards to which of these categories they belong to (see Table S1 and S2) and scUTRboot users can choose to exclude specific categories when performing differential isoform expression testing. As suggested, we have now clarified this information in the manuscript text (**lines 178, 179**).*

#Minor 4: Suppl Fig S5D: Is it unexpected that "Distal PAS Score" (which is APARENT2 score?) promotes overall differential shortening upon most pathway perturbations? I

would have naively guessed that a stronger distal PAS in general would be associated with lengthening.

Response: Figure S5d (now Fig. S6f) examines how various gene features correlate with changes in DUL across our groups of gene perturbations. The reviewer correctly notes that we find that expression of long 3'UTR isoforms derived from the use of strong PAS is reduced after depletion of 10/18 of the protein clusters examined.

The reason for the correlation of PAS quality (which is indeed APARENT2 score) and isoform expression may lie in the fact that CS with the highest PAS quality metrics often employ auxiliary mechanisms to promote cleavage. Although not essential for the processing reaction, these mechanisms may greatly enhance their kinetics. Our analysis identified many of these auxiliary factors as regulators of 3'UTR isoform expression and their knockdown “weakens” stronger sites disproportionately. For example, NUDT21, a component of the CFIm complex (cluster 1) that is recruited through cognate binding motifs close to the PAS hexamer, can strongly enhance mRNA cleavage. High-scoring PAS sites usually harbor NUDT21 binding sites, while low-scoring ones do not. Moreover, most functional NUDT21 binding sites are associated with distal CS, making these sites more competitive under normal conditions. Hence, NUDT21 knockdown preferentially causes shortening in genes where the distal sites have NUDT21 binding sites which coincides with high PAS scores. Similarly, the sequence context surrounding the CS could further improve isoform expression by coupling 3' end cleavage and mRNA export^{1,2}. Since long isoforms are more dependent on dedicated export pathways the depletion of export factors may lead to preferential shortening in genes that usually employ these mechanisms.

We added new text to lines 365-375 to clarify.

Notably, proximal PAS quality metrics were also correlated with several perturbation clusters. However, of the 13 perturbation clusters whose response was significantly correlated with proximal PAS scores, only 6 were also correlated with distal PAS scores. This suggests that proximal and distal CS may have evolved to be regulated by distinct molecular mechanisms.

Reviewer #3 (Remarks to the Author):

In this paper by Fansler et al from the Mayr lab, the authors have created a pipeline (scUTRquant) for the identification of human and mouse cleavage sites (CS) from very large amounts of single cell RNA-seq data (>100 datasets from each of mouse and human), resulting in the identification of ~40% more CS than are present in GENCODE. They conclude that most of the CS not present in GENCODE are used in only very few cell types and that most genes have only 1 or 2 CS that are used in the vast majority of cell types. They then examine single cell perturb-seq data published last year by the Weissman lab, focusing on CRISPRi-generated knock-downs of 2134 essential genes in order to compare effects on gene expression and CS choice. One central conclusion is that effects on CS choice are as common as those on gene expression. Another is that effects on gene expression and CS choice are mostly independent of each other, indicating that CS choice mostly affects processes other than gene expression (e.g. mRNA localization). Finally, by clustering the knockdowns that affect CS choice for similar gene groups, they identify 18 groups of gene that differ in the groups of genes whose CS choice they affect. They also conclude that effects on CS choice mostly affect the proximal CS, even

though CFI, which has the biggest effect, affects distal sites. There is much that is praiseworthy about this paper, especially its completeness for the phenomena that are described. On the other hand, the impact of the paper will be somewhat limited, both because the independence of gene expression and CS choice has already been described in multiple papers based on less complete data, and because the analysis of the perturb-seq data is limited by its lack of mechanistic insight.

We thank the reviewer for spending time with our manuscript and for the constructive comments. We would like to clarify that the pipeline used to identify CS from MWS data and generate the CS annotation did not use scUTRquant, but instead is a custom reprocessing of MWS data (see Methods). The scUTRquant pipeline does not identify CS, but rather is an optimized pipeline to quantify isoforms in a given annotation (either ours or custom ones generated with the txcutr tool).

Major points:

1. The authors point out that the CS they identify that are not present in GENCODE are generally found in PolyA-DB v3.2 and PolyASite 2.0, which means that most of the “new” CS they identify are not really new, only not present in GENCODE. Therefore, what is really new is the observation that most of the CS in the existing databases are rarely used (i.e. used in only a very few cell types). This needs to be made more clear.

*Response: We agree with the reviewer that the CS we have labeled as “MWS-only” sites (see Figures 1b-e) had been categorized as “novel” relative to the GENCODE annotation. This does not preclude that these sites have previously been identified by one of the dedicated 3' end atlases before. Specifically, after excluding GENCODE-only CS, 96% of the CS listed in our human annotation match with an entry in either PolyA-DB v3.2 or PolyASite 2.0 within 50 nts. Instead of writing ‘novel’ sites, we now use the more correct term MWS-only **throughout the manuscript** to better convey the fact that comparisons were made relative to GENCODE-provided 3' end annotations.*

Our annotation project has been strongly guided by its intended use in conjunction with our scUTRquant pipeline. With our work, we want to address a scientific audience that is not necessarily familiar with specialized 3' end databases. We acknowledge that researchers from the APA field are familiar with these databases and usually use them. However, with our method, we want to reach researchers that currently do not care about alternative 3'UTRs. We focused on GENCODE as the primary comparison because it remains the go-to resource for many researchers. We think that scientists performing scRNA-seq experiments for 3'UTR isoform quantification want to use a reliable resource and would require a comparison to the most commonly used alternative. With this in mind, we provide the reader with evidence for how certain performance metrics, such as the discovery of genes with differential 3'UTR length, can be improved using our curated database instead of GENCODE (see Fig. 3h and Fig. S4e).

However, even beyond the scUTRquant application, we think that our 3' end annotation provides a valuable resource for many scientists. In contrast to previous annotation projects, we intentionally decided to not call CS solely based on the technical detection limits but rather based on rationally designed criteria. This is important because rising sequencing depth has made it possible to record ever more unique 3' ends. Unfortunately, the resulting increase in 3' end annotations has yielded less useful resources for scientists from adjacent fields. For example, when working with data from current scRNA-seq data for isoform quantification,

overall data quality and interpretability may decrease when the potential for ambiguous read assignment increases. We have performed a level of data curation that has not been applied before to exclude unlikely CS while maintaining a comprehensive catalog of 3' ends. We also added labels for “major” and “minor” CS to our annotation which can be used to easily distinguish canonical from rare isoforms, respectively. Since we have used more than twice as many 3' end mapping reads than any previous annotation project, while also considering biological diversity across hundreds of cell types, we are confident that our curated atlas provides high value to a wider scientific community.

2. As the authors themselves point out, the independence of gene expression effects from CS effects, which is actually the title of the paper, has already been described for several specific cell types by papers from multiple labs, including the Mayr group (e.g. references 2, 23, and 24 from 2013 and 2014). This manuscript essentially generalizes that finding to comparisons of essentially any cell types.

Response: We agree that our conclusion is in line with previous studies which have been referenced in the present manuscript. However, we do disagree about the importance of this observation. Our motivation for the large-scale analysis presented here was the following. Although the independent relationship between gene expression and 3'UTR isoform expression was published by several different labs, the fact that only a limited set of conditions was analyzed led many researchers to conclude that the independence is probably restricted to a few conditions and will likely not be found in the majority of conditions. With our large-scale analysis, this argument can no longer be made, as it allows us to generalize the claim of independence of 3'UTR isoform and gene expression beyond specific contexts.

Moreover, we think that our rigorous and comprehensive analysis was needed because, up to this day, newly published papers continue to highlight the predominant impact of APA on gene expression³⁻⁶. We see this as evidence that the common understanding on this topic has not changed, potentially encouraging misleading claims and biased data interpretation.

We also believe that the widespread assumption that isoform expression is tightly related to total gene expression has limited the efforts to collect more isoform-resolved expression data in the past. Therefore, the finding that gene and isoform expression provide non-redundant information is central to our aim to encourage the broader scientific community to adopt isoform-specific quantification methods to gain overall more information.

3. The 18 gene groups that affect CS choice encompass a total of 836 genes out of the 2134 genes whose knockdowns were examined. That amounts to ~40% of the genes. On the one hand, that seems remarkable, because it is hard to believe that so many proteins could directly influence CS choice. However, these are all essential proteins, and they either affect cell cycle or essentially every aspect of the gene expression pathway (transcription [2 groups], chromatin, splicing [2 groups], CPA, mRNA export, translation [3 groups]). It is obvious that most of these effects must be indirect, perhaps very indirect. In fact, it is possible that none of the effects are direct except those of the cleavage factors themselves. Because of this lack of mechanistic insight, all we learn is that seriously messing up the cell almost always affects cleavage.

Response: The reviewer makes two points: (1) In the opinion of the reviewer, only cleavage factors can be considered direct regulators of differential 3'UTR isoform expression. (2) The

reviewer is concerned that many genes assigned to our APA regulator set may not be involved in APA regulation at all but may cause phenotypes associated with general cell stress.

Our pipeline analyzes steady-state transcript expression levels and their abundance is determined by both biogenesis and decay. Therefore, in addition to the cleavage factors that mostly influence CS choice, many other factors impact 3'UTR isoform expression. Indeed, among the APA regulator set of 836 genes, we found an enrichment of factors involved in all processes that regulate an mRNA's life cycle. One of the main conclusions of our analyses is that, despite originating from the same gene, differential 3'UTR isoform expression regulation can occur at various processing stages, including through posttranscriptional regulation. We added **new text to lines 376-383** to clarify.

Almost all clusters within the APA regulator set comprise genes involved in processes like mRNA transcription, processing, translation and decay. When performing gene ontology analysis with the APA regulator gene set (836 genes) over all analyzed essential genes (2,057 genes), we find that "RNA binding" (GO:0003723) is the top-ranked term (1.87×10^{-28} FDR-corrected p-value). Therefore, a large fraction of proteins encoded by the APA regulator gene set can directly interact with mRNA. We added **new text at lines 290-298** and added the GO analysis on the APA regulators to **Table S7**. Also see **Figure 1a below**. In contrast, there is a significant de-enrichment of terms related to intracellular transport, DNA binding and actin binding. Our analysis suggests that knock downs of essential genes related to these important functions do not elicit specific APA pattern changes.

In addition, we find that there is no correlation between the number of DUL genes and the genes' impact on cell fitness (**Figure 1b below**). Together, we think it is unlikely that cell stress is a significant source of false-positives in our gene perturbation analysis.

Figure 1: APA regulator gene set is enriched in genes with functions in RNA processing.

a) Enrichment (left) and depletion (right) of molecular function terms associated with the APA regulator set (836 genes) over all analyzed essential genes (2,057 genes).

b) Dot plot showing lack of correlation between the number of significant DUL genes per cluster and the genes' mean effect on cell fitness. Cell fitness was assessed with the Chronos score previously determined in K562 cells⁷.

4. Another downside of the analysis of the perturb-seq data is that it is never clear whether it is actually CS choice that is being affected. No matter which one of the 18 groups is being considered (with the exception of the cleavage factors themselves), one cannot tell whether the effect is on differential stability of the mRNA isoforms or else on CS choice.

*Response: Both mRNA production and degradation are important determinants of steady-state isoform expression levels. A major insight of our study is that the two processes are tightly linked. We observed a significant correlation between estimated isoform half-life and isoform expression changes across many perturbation clusters, especially among those that contain proteins primarily involved in transcriptional processes (see Fig. 5e and S6f). Isoform half-life correlates significantly with perturbation clusters 1, 2, 11-13, and 15-18 (which all contain factors involved in transcription and mRNA processing), but it does not correlate with clusters containing cytoplasmic factors, including the ribosome. This strongly indicates that a large part of mRNA degradation occurs co-transcriptionally and during mRNA maturation and export^{7,8} and cannot be cleanly separated from biogenesis processes. We added this paragraph to the manuscript **text at lines 376-383** to clarify.*

It is beyond the scope of this study to provide mechanistic insights into how much each perturbation cluster acts through modulation of isoform-specific production versus degradation. A thorough disentanglement of these processes on the isoform level across thousands of perturbations would require new experimental approaches alongside sophisticated data analysis methods.

5. It is not obvious that the 18 gene groups that affect CS choice are really distinct from one another. For example, the gene groups affected by splicing group I, splicing group II, and transcription group I all look similar. Also, the effects of the tRNA synthesis group, the DNA replication and repair group, transcription group II, CPSF, and the nuclear exosome look quite similar. Finally, the three translation groups look similar.

Response: We agree with the reviewer's assessment that some perturbation clusters exhibit similar patterns across target genes. As expected, the clusters containing genes involved in the same processes, such as splicing or translation, show the highest degree of similarity between them and we present data on their respective similarities in Fig. 5b.

The exact composition and size of perturbation clusters are the result of unsupervised clustering using user-defined parameters such as the minimum cluster size (see methods). Therefore, the number of clusters could be higher or lower to resolve either more or less gradual differences between their effects. The question is therefore, whether at the chosen level of resolution, the distinctions between closely related clusters still represent meaningful biological differences. As a representative example, we provide more detailed data describing the differences between the mRNA splicing clusters I and II for the reviewer (not part of the manuscript).

We agree that the differences between the two splicing clusters appear to be more quantitative rather than qualitative in nature. For example, 445 of 721 genes with significant DUL changes in splicing cluster I are also regulated by splicing cluster II, and no genes are oppositely regulated (see Figure 2a below). However, a subset of multi-UTR genes are distinctly affected by one but not the other perturbation cluster (see Figure 2a, 2b below). Intriguingly, we find that even between these two clusters, the components of several validated protein-protein complexes involved in splicing preferentially fall into one of the two clusters (see Figure 2c below). Among the evaluated gene features, we observe that the predicted cleavage probability of the distal CS significantly correlated with DUL effects for mRNA splicing cluster II but not mRNA splicing cluster I (Fig. S6f). Together, these findings argue that the differences between splicing clusters I and II are more likely to represent actual biological differences rather than technical differences related to the knock down efficiency.

Figure 2: Comparison of two perturbation clusters related to mRNA splicing. **a)** Comparison of DUL changes in mRNA splicing cluster I (rows) and cluster II (columns) for all multi-UTR genes that were tested in both conditions ($N = 3,326$ genes). **b)** Boxplot showing the average dWUI of multi-UTR gene groups from a) relative to non-targeting controls. **c)** Cluster membership of constituents of splicing-related protein complexes.

Minor point: there are some issues with the reference list. The first citation for a review on CPA only mentions one from the Mayr lab, and there are many other good ones. This is followed by the citation only of a Mayr lab paper showing how many genes use APA, and again there are others. Finally, there are references in the list that are missing various things (e.g. references 27, 28, 33).

Response: Previous APA reviews had been cited in the first paragraph of the paper but have been moved to the top upon request (ref. 1, 2). In the revised manuscript, more papers were cited for the widespread use of APA (ref. 4-6). We would like to thank the reviewer for pointing out the issues related to formatting of some of our references. These reference entries have now been corrected.

References

- 1 Muller-McNicoll, M, Botti, V, de Jesus Domingues, AM, Brandl, H, Schwich, OD, Steiner, MC, Curk, T, Poser, I, Zarnack, K & Neugebauer, KM. SR proteins are NXF1 adaptors that link alternative RNA processing to mRNA export. *Genes Dev* **30**, 553-566, doi:10.1101/gad.276477.115 (2016).
- 2 Li, J, Querl, L, Coban, I, Salinas, G & Krebber, H. Surveillance of 3' mRNA cleavage during transcription termination requires CF IB/Hrp1. *Nucleic Acids Res* **51**, 8758-8773, doi:10.1093/nar/gkad530 (2023).
- 3 Khajuria, DK, Nowak, I, Leung, M, Karuppagounder, V, Imamura, Y, Norbury, CC, Kamal, F & Elbarbary, RA. Transcript shortening via alternative polyadenylation promotes gene expression during fracture healing. *Bone Res* **11**, 5, doi:10.1038/s41413-022-00236-7 (2023).
- 4 Mittleman, BE, Pott, S, Warland, S, Zeng, T, Mu, Z, Kaur, M, Gilad, Y & Li, Y. Alternative polyadenylation mediates genetic regulation of gene expression. *eLife* **9**, doi:10.7554/eLife.57492 (2020).
- 5 Venkat, S, Tisdale, AA, Schwarz, JR, Alahmari, AA, Maurer, HC, Olive, KP, Eng, KH & Feigin, ME. Alternative polyadenylation drives oncogenic gene expression in pancreatic ductal adenocarcinoma. *Genome Res* **30**, 347-360, doi:10.1101/gr.257550.119 (2020).
- 6 de Prisco, N, Ford, C, Elrod, ND, Lee, W, Tang, LC, Huang, KL, Lin, A, Ji, P, Jonnakuti, VS, Boyle, L, Cabaj, M, Botta, S, Öunap, K, Reinson, K, Wojcik, MH, Rosenfeld, JA, Bi, W, Tveten, K, Prescott, T, Gerstner, T, Schroeder, A, Fong, CT, George-Abraham, JK, Buchanan, CA, Hanson-Khan, A, Bernstein, JA, Nella, AA, Chung, WK, Brandt, V, Jovanovic, M, Targoff, KL, Yalamanchili, HK, Wagner, EJ & Gennarino, VA. Alternative polyadenylation alters protein dosage by switching between intronic and 3'UTR sites. *Science advances* **9**, eade4814, doi:10.1126/sciadv.ade4814 (2023).
- 7 Dempster, JM, Boyle, I, Vazquez, F, Root, DE, Boehm, JS, Hahn, WC, Tsherniak, A & McFarland, JM. Chronos: a cell population dynamics model of CRISPR experiments that improves inference of gene fitness effects. *Genome biology* **22**, 343, doi:10.1186/s13059-021-02540-7 (2021).
- 8 Smalec, BM, Ietswaart, R, Choquet, K, McShane, E, West, ER & Churchman, LS. Genome-wide quantification of RNA flow across subcellular compartments reveals determinants of the mammalian transcript life cycle. *bioRxiv*, 2022.2008.2021.504696, doi:10.1101/2022.08.21.504696 (2022).

REVIEWER COMMENTS

Reviewer #1 (Remarks to the Author):

In the manuscript by Fansler et.al, the authors presented a systematic analysis of 3'UTR/PAS profile based on single cell RNA-seq data. First, they took advantage of microwell-seq data of 206 human and mouse cell types and annotated over 53,000 and 45,000 PAS sites in human and mouse, respectively. Based on the no. of cell type where these sites are used, the PAS sites could be characterized into major and minor ones. Intriguingly, most of genes have used only one or two major sites, which is in contrast to current PAS annotation which suggested that most genes used >5 PAS sites. Then, the authors developed a computational pipeline, scUTR quant for 3'UTR isoform quantification based on scRNA-seq data and validated its performance by comparing their results with those from bulk 3'seq data. Finally, by applying their tool to scRNA-seq data from cell differentiation, different cell types as well as genetic perturbation, they discovered 1) changes in UTR isoform usage is as widely as those in gene expression, but the two hit different gene sets. This resource expanded the commonly used Genecode annotation by 40%. plate-based high-throughput method to quantify ribosome-associated RNAs. The so-termed riboPLATE-seq is an extension of PLATE-seq, a protocol previously developed in the same lab. The authors then combined the two methods to estimate translation efficiency in cells treated with different kinase inhibitors.

Overall, the method described in the manuscript represented a rather limited technical advance. Moreover, the authors did not sufficiently evaluate the performance. Finally, the analysis of their data derived from different drug treatments is superficial and does not provide novel mechanistic understanding of translational regulation.

More specific points are listed below,

1. Comparing to Gencode annotation, what is the distribution of usage of novel sites versus common sites?
2. It would be interesting if the authors provide a figure/table summarizing the distribution of genes using more than one PAS site ($\geq 10\%$ usage) simultaneously in the different no. of cell types. This can reflect the APA regulation within the same cell type.
3. When GE and UTR isoform were compared between different samples, the two were often regulated in different set of genes. Have the authors checked whether GE was more likely changed for single UTR genes, in particular for the perturbation of core CPA machinery?
4. Since scUTRquant has limitation in distinguishing PAS sites in vicinity, how will this affect their conclusion, including the overlap of change in between GE and 3'UTR isoform, the clustering of gene perturbation etc.?
5. In all the perturbation experiments, are there any APA switch events, i.e. the switch of dominant 3'UTR isoforms? If so, what is the distribution of their proportions?
6. Change in RNA abundance and isoform usage could be due to either their biogenesis or decay. They should perform at least one perturbation for each of the 18 clusters and use bulk 3'seq to validate their findings, and then use 4sU labeling or transcription inhibition to distinguish the effects between processing and differential decay.
7. To a certain extent, the usage of different PAS sites is a competitive process, similar as alternative splicing. Why in figure 5d, most of 3'UTR isoform changes do not belong to the middle panel?

Reviewer #2 (Remarks to the Author):

Summary: Fansler et al constructed a comprehensive and cell type-resolved cleavage site (CS) annotation in human and mouse by mapping reads overlapping 3' cleavage sites from >200 Microwell-seq (MWS) data (collected from the Human Cell Landscape and Mouse Cell Atlas projects). The authors performed a number of post-processing steps to reduce artifacts (e.g. internal priming) and compared their measurements to orthogonal tools (e.g. APARENT2) and datasets (e.g. conservation scores, overlap with GENCODE-, PolyADB- and PolyASite annotations, etc.) for quality control. The authors demonstrated the utility of their new annotation for 3' isoform quantification in scRNA-seq data, with applications ranging from erythroid differentiation to analyzing scPerturb-seq data. By combining their 3' isoform quantification with network- and pathway analyses, the authors make pathway-specific observations about overall trends in global UTR shortening or lengthening upon perturbation.

Impact: Overall I found this to be a nicely executed study that resulted in a much-needed consensus annotation/quantification of cleavage sites and their usage in diverse human cell types. The axis of cell type-specificity has been largely absent from earlier pA site atlas efforts so I expect this will be of wide use for the research community. The authors perform a lot of QC to verify their annotation. Additionally, the authors spend a good deal of effort using their annotation to analyze a large perturb-seq dataset and draw interesting conclusions about general shortening trends across different regulatory pathways, and quantify the (surprising, but increasingly more accepted) observation that gene expression and shortening is less coupled than what was earlier believed.

I have a few comments and suggestions regarding some of the analyses. After these have been addressed, I recommend accepting the manuscript for publication.

Major comments: I have no major comments / criticisms.

Minor comments:

Minor #1: In the paragraph starting on Line 112 (and associated Fig. 1g), one comparison that I feel is missing is that between "CS usage" (as authors define it: frequency of being used at all across cell types) and isoform abundance proportion (i.e. how much the CS is used in comparison to competing proximal or distal CS in the same gene and cell type). I suggest adding a (simple) comparison of some sort (e.g. scatter plot or histograms) comparing these metrics. Otherwise it's not clear to me whether the cell type-common cleavage sites are lowly-used/weak signals or strong/canonical signals (though I'm guessing it's the latter).

Minor #2: It might be prudent to add some hypothesis or reason for why the replicate concordance is so much better in scRNA-seq data than in bulk 3' sequencing data, given the same annotation for quantification. Is it just a function of read depth?

#Minor 3: Regarding the paragraph starting at Line 166 (and associated Fig. 3a and onward) : Does UTRome (used for scUTRquant) include intronic pA annotations, or only 3' UTR CS events? It was not immediately clear to me (though in the next paragraph the authors mention IPA isoforms so I assume yes). It may be worth clarifying this in the paragraph at Line 166.

#Minor 4: Suppl Fig S5D: Is it unexpected that "Distal PAS Score" (which is APARENT2 score?) promotes overall differential shortening upon most pathway perturbations? I would have naively guessed that a stronger distal PAS in general would be associated with lengthening.

Reviewer #3 (Remarks to the Author):

In this paper by Fansler et al from the Mayr lab, the authors have created a pipeline (scUTRquant) for the identification of human and mouse cleavage sites (CS) from very large amounts of single cell RNA-seq data (>100 datasets from each of mouse and human), resulting in the identification of ~40% more CS than are present in GENCODE. They conclude that most of the CS not present in GENCODE are used in only very few cell types and that most genes have only 1 or 2 CS that are used in the vast majority of cell types. They then examine single cell perturb-seq data published last year by the Weissman lab, focusing on CRISPRi-generated knock-downs of 2134 essential genes in order to compare effects on gene expression and CS choice. One central conclusion is that effects on CS choice are as common as those on gene expression. Another is that effects on gene expression and CS choice are mostly independent of each other, indicating that CS choice mostly affects processes other than gene expression (e.g. mRNA localization). Finally, by clustering the knockdowns that affect CS choice for similar gene groups, they identify 18 groups of genes that differ in the groups of genes whose CS choice they affect. They also conclude that effects on CS choice mostly affect the proximal CS, even though CFI, which has the biggest effect, affects distal sites. There is much that is praiseworthy about this paper, especially its completeness for the phenomena that are described. On the other hand, the impact of the paper will be somewhat limited, both because the independence of gene expression and CS choice has already been described in multiple papers based on less complete data, and because the analysis of the perturb-seq data is limited by its lack of mechanistic insight.

Major points:

1. The authors point out that the CS they identify that are not present in GENCODE are generally found in PolyA-DB v3.2 and PolyASite 2.0, which means that most of the "new" CS they identify are not really new, only not present in GENCODE. Therefore, what is really new is the observation that most of the CS in the existing databases are rarely used (i.e. used in only a very few cell types). This needs to be made more clear.
2. As the authors themselves point out, the independence of gene expression effects from CS effects, which is actually the title of the paper, has already been described for several specific cell types by papers from multiple labs, including the Mayr group (e.g. references 2, 23, and 24 from 2013 and 2014). This manuscript essentially generalizes that finding to comparisons of essentially any cell types.
3. The 18 gene groups that affect CS choice encompass a total of 836 genes out of the 2134 genes whose knockdowns were examined. That amounts to ~40% of the genes. On the one hand, that seems remarkable, because it is hard to believe that so many proteins could directly influence CS choice. However, these are all essential proteins, and they either affect cell cycle or essentially every aspect of the gene expression pathway (transcription [2 groups], chromatin, splicing [2 groups], CPA, mRNA export, translation [3 groups]). It is obvious that most of these effects must be indirect, perhaps very indirect. In fact, it is possible that none of the effects are direct except those of the cleavage factors themselves. Because of this lack of mechanistic insight, all we learn is that seriously messing up the cell almost always affects cleavage.
4. Another downside of the analysis of the perturb-seq data is that it is never clear whether it is actually CS choice that is being affected. No matter which one of the 18 groups is being considered (with the exception of the cleavage factors themselves), one cannot tell whether the effect is on differential stability of the mRNA isoforms or else on CS choice.
5. It is not obvious that the 18 gene groups that affect CS choice are really distinct from one another. For example, the gene groups affected by splicing group I, splicing group II, and transcription group I all look similar. Also, the effects of the tRNA synthesis group, the DNA replication and repair group, transcription group II, CPSF, and the nuclear exosome look quite similar. Finally, the three translation groups look similar.

Minor point: there are some issues with the reference list. The first citation for a review on CPA only mentions one from the Mayr lab, and there are many other good ones. This is followed by the citation only of a Mayr lab paper showing how many genes use APA, and again there are others. Finally, there are references in the list that are missing various things (e.g. references 27, 28, 33).